# AdaptDel: Adaptable Deletion Rate Randomized Smoothing for Certified Robustness

**Zhuoqun Huang**
University of Melbourne
zhuoqun@unimelb.edu.au

**Neil G. Marchant**
University of Melbourne
nmarchant@unimelb.edu.au

**Olga Ohrimenko**
University of Melbourne
oohrimenko@unimelb.edu.au

**Benjamin I. P. Rubinstein**
University of Melbourne
brubinstein@unimelb.edu.au

## Abstract

We consider the problem of certified robustness for sequence classification against edit distance perturbations. Naturally occurring inputs of varying lengths (e.g., sentences in natural language processing tasks) present a challenge to current methods that employ fixed-rate deletion mechanisms and lead to suboptimal performance. To this end, we introduce AdaptDel methods with adaptable deletion rates that dynamically adjust based on input properties. We extend the theoretical framework of randomized smoothing to variable-rate deletion, ensuring sound certification with respect to edit distance. We achieve strong empirical results in natural language tasks, observing up to 30 orders of magnitude improvement to median cardinality of the certified region, over state-of-the-art certifications.

## 1 Introduction

Recent advancements in machine learning have led to significant improvements across many domains, however the fragility of models to benign noise [1] and adversarial perturbations [2] continues to undermine model reliability. While empirical defenses have shown promising results in enhancing model robustness [3–8], they often lack guarantees when facing adaptive adversaries capable of exploiting the model's defense strategies. Consequently, certified defenses have emerged as a promising alternative, providing robustness guarantees against arbitrary attacks within a defined threat model [9–13]. Among certified defenses, randomized smoothing [14, 15] has gained widespread recognition for its scalability to large models and strong robustness guarantees.

Sequence prediction tasks, fundamental to natural language processing, bioinformatics, and time series analysis, present unique challenges for certified robustness. Established threat models, such as bounded $\ell_p$ perturbations, are fundamentally incompatible with variable-length, discrete sequences. Moreover, the prevailing additive noise-based smoothing mechanisms are ill-defined for discrete data. To address these challenges, smoothing mechanisms have been proposed that additively perturb and permute sequences in embedding space [16] or delete sequence elements [17]. The deletion-based mechanism provides certificates under bounded edit distance perturbations, whereas the perturbation/permutation-based mechanisms provide certificates in embedding space and typically fail to cover edit distance perturbations of any magnitude [18, Appendix F]. This focus on variable-length inputs is a key differentiator from many certified defenses in computer vision, where fixed-size inputs are the norm and length-dependent adaptation is less critical.

While deletion-based mechanisms have shown promise, one area where existing mechanisms fall short is their use of a fixed deletion rate for all inputs. This one-size-fits-all approach disregards

39th Conference on Neural Information Processing Systems (NeurIPS 2025).

potential variations in sequence length, complexity, or domain-specific characteristics. For instance, Huang et al. [18] observed that longer text sequences can typically tolerate higher deletion rates without compromising predictive accuracy. This insight suggests a fixed deletion rate may lead to suboptimal trade-offs between robustness and performance, particularly for inputs that could benefit from more nuanced treatment.

The potential of input-dependent smoothing for discrete sequences remains largely unexplored. Prior work has considered Gaussian smoothing with input-dependent noise for $\ell_p$ certification of real-valued inputs. Other work has taken an ad hoc approach, where Cohen et al.'s certificate for uniform Gaussian noise is corrected iteratively at test time [19, 20]. However, this results in a model where each output depends on the chain of previous outputs, which complicates interpretability, may lead to error propagation and increases computational complexity and memory requirements. A more rigorous approach was taken by Súkeník et al. [21], who derived a certificate for input-dependent Gaussian noise.

In this paper, we extend input-dependent smoothing to discrete sequences, where a new analysis is required to obtain sound edit distance certificates. We begin by introducing a general framework for deletion smoothing with input-dependent deletion rates, which provides bounds on the smoothed model's confidence scores under perturbations. Building on these bounds, we propose two methods that support efficient edit distance certification: AdaptDel, which uses a length-dependent deletion rate, and AdaptDel+, which further optimizes the deletion rate using input binning and empirical calibration. Our contributions are summarized as follows:

- We develop a theoretical foundation for variable deletion rates, enabling robust smoothing mechanisms that adapt to input length.

- We propose AdaptDel and AdaptDel+, two novel adaptive smoothing techniques, which enable computationally efficient certified robustness while maintaining high accuracies.

- We evaluate our methods on four natural language processing tasks with varying input sizes, demonstrating that AdaptDel and AdaptDel+ achieve stronger robustness on all datasets at the same certified accuracy, with little to no degradation in clean accuracy: for example, we observe up to 30 orders of magnitude improvement to median cardinality of the certified region.

## 2   Preliminaries

**Task**    Let $\mathcal{X} = \Omega^*$ denote the set of finite-length sequences with tokens drawn from vocabulary $\Omega$. For a sequence $\boldsymbol{x} \in \mathcal{X}$, let the length be denoted by $|\boldsymbol{x}|$, and let the $k$-th token be denoted by $x_k$. We consider a sequence classification task, where a model $f$ predicts the class $y \in \mathcal{Y}$ of input sequence $\boldsymbol{x}$.

Though our methods apply to generic sequence classification, we focus on text for concreteness, where text is mapped to a sequence using a tokenizer. For instance, in topic classification, the input text might be a news article and the possible classes are topics like "politics" and "sports" [22].

**Edit Distance**    While it is common to consider $\ell_p$-based threat models for the image domain, textual data is subject to more structured perturbations, such as edit-based perturbations [23–25]. Given two sequences $\boldsymbol{x}, \bar{\boldsymbol{x}} \in \mathcal{X}$, and a subset of edit operations $\mathsf{o} \subseteq \{\mathsf{del}, \mathsf{ins}, \mathsf{sub}\}$, we define the edit distance $\mathrm{dist}_\mathsf{o}(\boldsymbol{x} \twoheadrightarrow \bar{\boldsymbol{x}})$ as the minimum number of edits required to transform $\boldsymbol{x}$ into $\bar{\boldsymbol{x}}$. We note that edit distance is not symmetric if $\mathsf{o}$ contains $\mathsf{del}$ without $\mathsf{ins}$ or vice versa, which explains our use of $\twoheadrightarrow$ to emphasize directionality. Although $\mathrm{dist}_\mathsf{o}$ is not necessarily a distance metric, this is not a problem for our analysis—our results hold regardless.

**Edit Distance Robustness**    Given a classifier $f$, we consider the problem of certifying its robustness under bounded edit distance perturbations. Specifically, we are interested in certifying that the classifier's prediction is invariant to any input perturbation of magnitude $r$ in edit distance,

$$\forall \bar{\boldsymbol{x}} \in B_r(\boldsymbol{x}; \mathsf{o}) : f(\boldsymbol{x}) = f(\bar{\boldsymbol{x}}), \tag{1}$$

where $B_r(\boldsymbol{x}; \mathsf{o}) := \{\bar{\boldsymbol{x}} \in \mathcal{X} : \mathrm{dist}_\mathsf{o}(\boldsymbol{x} \twoheadrightarrow \bar{\boldsymbol{x}}) \leq r\}$ denotes the edit distance ball of radius $r$ centered on $\boldsymbol{x}$.

As is typical for randomized smoothing, we will develop mechanisms that produce a randomized radius $r$ given input sequence $\boldsymbol{x}$, such that with high probability $1 - \alpha$, this radius is a valid certificate at $\boldsymbol{x}$. We achieve this certification against general edit distance using a smoothing mechanism based exclusively on deletions, which we introduce and justify in Section 3.

# 3 Adaptable Deletion Certification

We now present our framework for deletion-based randomized smoothing with a deletion rate that adapts depending on the input. Our framework generalizes fixed-rate randomized deletion [17], achieving edit distance certificates with superior robustness-accuracy trade-offs over a wider range of inputs. To begin, we review randomized smoothing in Section 3.1 and describe our variable-rate deletion mechanism in Section 3.2. Next, in Section 3.3 we obtain bounds on the smoothed classifier's scores under input perturbations, where we make no assumptions about the deletion rate's dependence on the input. Although the adaptive setting makes the analysis significantly more complex, we obtain bounds that can be evaluated algorithmically by solving a bounded knapsack problem. Finally, in Section 3.4 we specialize our framework to length-dependent deletion rates, which allows us to efficiently compute an edit distance certificate.

## 3.1 Randomized Smoothing

While various techniques have been developed for certified robustness, randomized smoothing [14] is a leading approach owing to its scalability to large models and the size of the robust radii it can achieve. Given a *base classifier* $f_{\mathrm{b}} : \mathcal{X} \to \mathcal{Y}$, and a *smoothing mechanism* $\phi : \mathcal{X} \to \mathcal{D}(\mathcal{X})$ that maps the input to a distribution of perturbed inputs, randomized smoothing defines a *smoothed classifier* $f : \mathcal{X} \to \mathcal{Y}$ such that

$$f(\boldsymbol{x}) \coloneqq \arg\max_{y \in \mathcal{Y}} p_y(\boldsymbol{x}; f_{\mathrm{b}}, \phi)$$

where $p_y(\boldsymbol{x}; f_{\mathrm{b}}, \phi)$ is the smoothed class probability $p_y(\boldsymbol{x}; f_{\mathrm{b}}, \phi) = \mathrm{Pr}_{\boldsymbol{z} \sim \phi(\boldsymbol{x})}[f_{\mathrm{b}}(\boldsymbol{z}) = y]$. The effectiveness of randomized smoothing heavily relies on the choice of smoothing mechanism. An ideal mechanism will achieve large robustness certificates in the desired threat model while minimizing the performance degradation.

## 3.2 Variable-Rate Deletion

We now present a smoothing mechanism that randomly deletes sequence tokens, where the rate of deletion varies as a function of the input sequence. Our mechanism generalizes the fixed-rate deletion mechanism proposed by Huang et al. [17].

We begin by defining notation to express the mechanism symbolically. Let $\boldsymbol{\epsilon} = (\epsilon_1, \dots, \epsilon_{|\boldsymbol{x}|})$ be a vector of deletion indicator variables for input text $\boldsymbol{x}$ containing $|\boldsymbol{x}|$ tokens, where $\epsilon_i = 1$ if the $i$-th token is to be retained and $\epsilon_i = 0$ if it is to be deleted. We slightly abuse notation and let $|\boldsymbol{\epsilon}| \coloneqq \sum_i \epsilon_i$ denote the number of tokens retained. The space of possible deletion indicators for input text $\boldsymbol{x}$ is denoted as $\mathcal{E}(\boldsymbol{x}) = \{0, 1\}^{|\boldsymbol{x}|}$. Our variable-rate deletion mechanism is parametrized by a deletion probability function $\psi : \mathcal{X} \to [0, 1]$, that maps an input $\boldsymbol{x}$ to a deletion probability shared across tokens. The probability mass function for $\boldsymbol{\epsilon}$ at input $\boldsymbol{x}$ takes the form $q(\boldsymbol{\epsilon}|\boldsymbol{x}) = \prod_i \psi(\boldsymbol{x})^{1 - \epsilon_i} (1 - \psi(\boldsymbol{x}))^{\epsilon_i}$, i.e., each deletion indicator is drawn i.i.d. from a Bernoulli distribution with probability $\psi(\boldsymbol{x})$. We write $\mathrm{apply}(\boldsymbol{x}, \boldsymbol{\epsilon})$ to denote the resultant sequence after deleting the tokens referenced in $\boldsymbol{\epsilon}$ from input text $\boldsymbol{x}$. The end result of this process is a random subsequence of $\boldsymbol{x}$, which we denote by $\phi_\psi(\boldsymbol{x})$, emphasizing the dependence on $\psi$.

Using this notation, we can express the smoothed class probability as a sum over the space of deletion indicator variables:

$$p_y(\boldsymbol{x}; f_{\mathrm{b}}, \phi_\psi) = \sum_{\boldsymbol{\epsilon} \in \mathcal{E}(\boldsymbol{x})} s_{f_{\mathrm{b}}}(\boldsymbol{\epsilon}, \boldsymbol{x}) \quad \text{with} \quad s_{f_{\mathrm{b}}}(\boldsymbol{\epsilon}, \boldsymbol{x}) = q(\boldsymbol{\epsilon}|\boldsymbol{x}) \mathbf{1}_{f_{\mathrm{b}}(\mathrm{apply}(\boldsymbol{x}, \boldsymbol{\epsilon})) = y}, \tag{2}$$

where we define $\mathbf{1}_A$ as the indicator function that returns 1 if the predicate $A$ is true and 0 otherwise. We conceal the dependence on $\phi_\psi$ and write $p_y(\boldsymbol{x}; f_{\mathrm{b}})$ where evident.

*Remark* 1 (On Smoothing vs. Certified Edits). Our smoothing mechanism uses only deletions, while our edit distance certificate (derived in the following sections) provides robustness against a broader

set of edits, including insertions and substitutions. This distinction is valid because the primary role of a smoothing mechanism is to ensure that nearby sequences under the certificate's distance measure produce statistically similar distributions of perturbed outputs. Since an edit distance ball contains sequences of varying lengths, the smoothing mechanism must be able to change the input's dimensionality. Deletion is a simple and effective operator for this purpose; by randomly removing tokens, it maps different source sequences to a common, lower-dimensional subsequence. While a mechanism incorporating insertions or substitutions could also be designed, deletion alone is sufficient to create the statistical overlap necessary for certification.

### 3.3 Analysis for Variable-Rate Deletion

We now turn our attention to deriving a certificate for randomized smoothing with variable-rate deletion, where we make no assumption about the functional form of the deletion rate $\psi$. Our key results are lower and upper bounds on the smoothed class probability $p_y(\bar{x}; f_{\mathrm{b}})$ at a neighboring input $\bar{x}$, that depend solely on the smoothed class probability at the original input $x$ and the size of the longest common subsequence between $x$ and $\bar{x}$. These bounds can be used to perform brute force certification, or to derive more certificates under simplifying assumptions on the functional form of the deletion rate $\psi$ (discussed in the next section).

The first step in our analysis is to identify a correspondence between deletions for neighboring inputs. Let

$$\sqsubseteq = \{\, (\epsilon, \epsilon') \in \mathcal{E}(x) \times \mathcal{E}(x) : \forall i, \epsilon_i \geq \epsilon_i' \,\}.$$

be a partial order on the space of deletion indicators $\mathcal{E}(x)$. We can then write $\epsilon \sqsubseteq \epsilon'$ if $\epsilon'$ can be obtained from $\epsilon$ by deleting additional tokens.

This allows us to define a set of deletions building on $\epsilon$:

$$\mathcal{E}(x, \epsilon) := \{\, \epsilon' \in \mathcal{E}(x) : \epsilon \sqsubseteq \epsilon' \,\},$$

which represents all possible deletions that extend the base deletion $\epsilon$ under the space of deletion indicators $\mathcal{E}$. We further define the subset of deletions that result in subsequences of a specific length:

$$\mathcal{E}^k(x, \epsilon^\star) := \{\epsilon \in \mathcal{E}(x, \epsilon^\star) \mid |\epsilon| = k\},$$

which represents the set of all equivalent edits achieving the same length $k$ by summing the deletion indicators.

The following result adapted from [17, Lemma 4] identifies a relationship between terms in Equation 2 for a pair of inputs $x, \bar{x}$ based on their longest common subsequence.

**Lemma 2** (17). *Let $z^\star$ be a longest common subsequence [26] of $\bar{x}$ and $x$, and let $\bar{\epsilon}^\star \in \mathcal{E}(\bar{x})$ and $\epsilon^\star \in \mathcal{E}(x)$ be any deletions such that $\mathrm{apply}(\bar{x}, \bar{\epsilon}^\star) = \mathrm{apply}(x, \epsilon^\star) = z^\star$. Then there exists a bijection $m \colon \mathcal{E}(\bar{x}, \bar{\epsilon}^\star) \to \mathcal{E}(x, \epsilon^\star)$ such that $\mathrm{apply}(\bar{x}, \bar{\epsilon}) = \mathrm{apply}(x, \epsilon)$ for any $\bar{\epsilon} \sqsupseteq \bar{\epsilon}^\star$ and $\epsilon = m(\bar{\epsilon})$. Furthermore, we have*

$$s_{f_{\mathrm{b}}}(\bar{\epsilon}, \bar{x}) = \frac{\bar{\psi}^{|\bar{x}|}}{\psi^{|x|}} \rho(\psi, \bar{\psi})^{|\epsilon|} s_{f_{\mathrm{b}}}(\epsilon, x),$$

*where we define $\psi := \psi(x)$, $\bar{\psi} := \psi(\bar{x})$ and $\rho(\psi, \bar{\psi}) = \frac{\psi(1-\bar{\psi})}{\bar{\psi}(1-\psi)}$ for conciseness.*

We can decompose the sum over edits in Equation 2 into two parts: a sum over edits in the set $\mathcal{E}(x, \epsilon^\star)$ building on $\epsilon^\star$ and a sum over the remaining edits not in this set. After invoking Lemma 2, we obtain the following relation between the smoothed class probability at $\bar{x}$ and $x$:

$$p_y(\bar{x}; f_{\mathrm{b}}) = \frac{\bar{\psi}^{|\bar{x}|}}{\psi^{|x|}} \sum_{\epsilon \in \mathcal{E}(x, \epsilon^\star)} \rho(\psi, \bar{\psi})^{|\epsilon|} s_{f_{\mathrm{b}}}(\epsilon, x) \quad + \sum_{\bar{\epsilon} \in \mathcal{E}(\bar{x}) \setminus \mathcal{E}(\bar{x}, \bar{\epsilon}^\star)} s_{f_{\mathrm{b}}}(\bar{\epsilon}, \bar{x}). \quad (3)$$

From here, we obtain a lower bound on the right-hand side of (3) by dropping the sum over $\mathcal{E}(\bar{x}) \setminus \mathcal{E}(\bar{x}, \bar{\epsilon}^\star)$ and deriving a lower bound on the sum over $\mathcal{E}(x, \epsilon^\star)$ that is the solution to a bounded knapsack problem. Later, we use this pairwise lower bound to obtain an edit distance certificate by considering the worst case $\bar{x}$ within a bounded edit distance from $x$.

**Lemma 3.** *Let $\boldsymbol{x}, \bar{\boldsymbol{x}} \in \mathcal{X}$ be a pair of inputs with a longest common subsequence (LCS) $\boldsymbol{z}^\star$ and let $\mu = p_y(\boldsymbol{x}; f_{\mathrm{b}})$. Define*

$$H^* = \begin{cases} \min_{h:\sum_{i=0}^{h} \mathcal{B}_i(|\boldsymbol{z}^\star|, \psi) \geq \mu - 1 + \psi^{|\boldsymbol{x}| - |\boldsymbol{z}^\star|}} h, & \psi \geq \bar{\psi}, \\ \max_{h:\sum_{i=h}^{|\boldsymbol{z}^\star|} \mathcal{B}_i(|\boldsymbol{z}^\star|, \psi) \geq \mu - 1 + \psi^{|\boldsymbol{x}| - |\boldsymbol{z}^\star|}} h, & \psi < \bar{\psi}, \end{cases}$$

*as a threshold on the number of tokens retained when editing $\boldsymbol{x}$, where $\mathcal{B}_k(n, p) := \binom{n}{k}(1-p)^k p^{n-k}$ is the Binomial pmf for $n$ trials with success probability $1 - p$. Then there exists a lower bound $\mathrm{lb}(\mu, \boldsymbol{x}, \bar{\boldsymbol{x}}, \psi) \leq p_y(\bar{\boldsymbol{x}}; f_{\mathrm{b}})$ such that:*

$$\mathrm{lb}(\mu, \boldsymbol{x}, \bar{\boldsymbol{x}}, \psi) = \frac{\bar{\psi}^{|\bar{\boldsymbol{x}}| - |\boldsymbol{z}^\star|}}{\psi^{|\boldsymbol{x}| - |\boldsymbol{z}^\star|}} \left( \sum_{i=l(H^*+1)}^{(1-l)(H^*-1)+l|\boldsymbol{z}^\star|} \mathcal{B}_i(|\boldsymbol{z}^\star|, \bar{\psi}) \right.$$

$$\left. + \mathcal{B}_{H^*}(|\boldsymbol{z}^\star|, \bar{\psi}) \left\lfloor \frac{c(\mu, |\boldsymbol{x}|, |\boldsymbol{z}^\star|, \psi, H^*)}{\mathcal{B}_{H^*}(|\boldsymbol{z}^\star|, \psi)} \right\rfloor_{\binom{|\boldsymbol{z}^\star|}{H^*}^{-1}} \right),$$

*where*

$$c(\mu, |\boldsymbol{x}|, |\boldsymbol{z}^\star|, \psi, H^*) = \mu - 1 + \psi^{|\boldsymbol{x}| - |\boldsymbol{z}^\star|} - \sum_{i=l(H^*+1)}^{(1-l)(H^*-1)+l|\boldsymbol{z}^\star|} \mathcal{B}_j(|\boldsymbol{z}^\star|, \psi),$$

*$l = \mathbf{1}_{\psi < \bar{\psi}}$ is a binary indicator and $\lfloor \cdot \rfloor_v := \lfloor \frac{\cdot}{v} \rfloor v$ is a gridded flooring operation.*

*Proof sketch.* We consider the case where $\psi \geq \bar{\psi}$, as the opposite case follows similar logic. Dropping the last term in Equation 3, we obtain the bound:

$$p_y(\bar{\boldsymbol{x}}; f_{\mathrm{b}}) \geq \frac{\bar{\psi}^{|\bar{\boldsymbol{x}}|}}{\psi^{|\boldsymbol{x}|}} \sum_{\boldsymbol{\epsilon} \in \mathcal{E}(\boldsymbol{x}, \boldsymbol{\epsilon}^\star)} \rho(\psi, \bar{\psi})^{|\boldsymbol{\epsilon}|} s_{f_{\mathrm{b}}}(\boldsymbol{\epsilon}, \boldsymbol{x}).$$

Since all factors are non-negative, we focus on lower bounding the sum over $\mathcal{E}(\boldsymbol{x}, \boldsymbol{\epsilon}^\star)$, which we denote by $\sigma(\bar{\boldsymbol{x}}; f_{\mathrm{b}})$. We lower bound $\sigma(\bar{\boldsymbol{x}}; f_{\mathrm{b}})$ by replacing the base model $f_{\mathrm{b}}$ by a worst-case plausible model

$$\sigma(\bar{\boldsymbol{x}}; f_{\mathrm{b}}) \geq \frac{\bar{\psi}^{|\bar{\boldsymbol{x}}|}}{\psi^{|\boldsymbol{x}|}} \min_{\bar{f} \in \mathcal{F}} \sum_{\boldsymbol{\epsilon} \in \mathcal{E}(\boldsymbol{x}, \boldsymbol{\epsilon}^\star)} \rho(\psi, \bar{\psi})^{|\boldsymbol{\epsilon}|} s_{\bar{f}}(\boldsymbol{\epsilon}, \boldsymbol{x})$$

where the set of plausible models satisfying the constraint in Lemma 5 is

$$\mathcal{F} = \left\{ \bar{f} \,\Big|\, \sum_{\boldsymbol{\epsilon} \in \mathcal{E}(\boldsymbol{x}, \boldsymbol{\epsilon}^\star)} s_{\bar{f}}(\boldsymbol{\epsilon}, \boldsymbol{x}) \geq \mu - 1 + \psi^{|\boldsymbol{x}| - |\boldsymbol{z}^\star|} = W \right\}.$$

The minimization problem can now be viewed as a bounded knapsack problem [27], where each $\boldsymbol{\epsilon}$ is treated as an item of size $i = |\boldsymbol{\epsilon}|$, with weight $w(i)$ and value $v(i)$. So for size $i$ at most $\binom{N}{i}$ items are available. We choose multiplicities $h(i) \leq \binom{N}{i}$ to minimize $\sum_{i=0}^{N} v(i)h(i)$ subject to $\sum_{i=0}^{N} w(i)h(i) \geq W$. Since the unit-value ratio $v(i)/w(i) = \rho(\psi, \bar{\psi})^i$ increases in $i$, the greedy solution orders by $i$ filling up to a critical $H^*$ where the cumulative weight would meet $W$, then takes as many items of size $H^*$ as possible without meeting the weight requirement. This yields a valid, though not necessarily tight, lower bound. □

By a similar argument, we also obtain an upper bound on $p_y(\bar{\boldsymbol{x}}; f_{\mathrm{b}})$ in Lemma 6 (deferred to Appendix A due to space constraints) that is the solution to a maximization problem viewed as a bounded knapsack problem [28]. The upper bound is not needed when computing a certificate in the style of Cohen et al. [14]. However, it is needed to compute certificates in the style of Lecuyer et al. [15] or for certification of top-k predictions [29].

We note that the bounds in Lemmas 3 and 6 can be used to compute an edit distance certificate by exhaustive search for arbitrary deletion functions (see Appendix B) with a constant factor improvement over querying the smoothed classifier directly. By making further assumptions on the deletion function, we are able to compute certificates more efficiently, as demonstrated in the next section.

## 3.4 Certification for Length-Dependent Deletion

We now turn our attention to the class of deletion functions $\psi(\boldsymbol{x})$ that depend on the input only via the length $|\boldsymbol{x}|$. To emphasize this restricted form of dependence on $\boldsymbol{x}$, we write $\psi(\boldsymbol{x}) = \psi(|\boldsymbol{x}|)$ in this section. For $\psi$ of this form, the lower and upper bounds on $p_y(\bar{\boldsymbol{x}}, f_{\mathrm{b}})$ in Lemmas 3 and 6 depend only on $p_y(\boldsymbol{x}, f_{\mathrm{b}})$, the sequence lengths $|\boldsymbol{x}|$ and $|\bar{\boldsymbol{x}}|$, and the length of the LCS $|\boldsymbol{z}^\star|$. We can therefore enumerate over all neighboring sequences $\bar{\boldsymbol{x}}$ within edit distance $r$ of $\boldsymbol{x}$ by enumerating over the number of edits $n_e$ of each type in $e \in \mathsf{o}$, such that the total number of edits $n_{\mathsf{del}} + n_{\mathsf{ins}} + n_{\mathsf{sub}}$ does not exceed $r$. The set containing the possible number of edits of each type at radius $r$ is defined as follows:

$$\mathcal{C}(\mathsf{o}, r) = \Big\{ (n_{\mathsf{del}}, n_{\mathsf{ins}}, n_{\mathsf{sub}}) \in \mathbb{N} \cup \{0\} :$$

$$n_{\mathsf{del}} + n_{\mathsf{ins}} + n_{\mathsf{del}} = r, n_{\mathsf{del}} \leq r \mathbf{1}_{\mathsf{del} \in \mathsf{o}}, n_{\mathsf{ins}} \leq r \mathbf{1}_{\mathsf{ins} \in \mathsf{o}}, n_{\mathsf{sub}} \leq r \mathbf{1}_{\mathsf{sub} \in \mathsf{o}} \Big\}.$$

We can now establish Algorithm 1 and Theorem 4 that certify the top prediction of a smoothed classifier with length-dependent deletion function $\psi$ using these insights. It follows the standard Monte Carlo estimation approach from prior work [14, 15]. Lines 1 and 2 estimate a lower confidence bound for the predicted class $y_1$ and an upper confidence bound for the runner up class $y_2$ following the approach of Lecuyer et al. [15] with a designated $\alpha$ confidence bound. We use $\hat{p}_y(\boldsymbol{x}; f_{\mathrm{b}}, \phi_\psi, \alpha)$ with superscript lb or ub to denote the Clopper-Pearson lower and upper confidence bound for the smoothed class probability $p_y$. This involves a Bonferroni correction (dividing $\alpha$ by 2) since a one-sided confidence bound is estimated for two-classes using the same sample drawn from $\phi_\psi(\boldsymbol{x})$. We circumvent the problem of needing to divide the confidence level by the number of classes by estimating the

---

**Algorithm 1** CERTIFY

**Require:** base classifier $f_{\mathrm{b}}$, input sequence $\boldsymbol{x}$, predicted class $y_1$, length-dependent deletion probability $\psi$, allowed edit operations $\mathsf{o}$, significance level $\alpha$

**Ensure:** maximum radius that can be certified
1: $t_1^{\mathsf{lb}} \leftarrow \hat{p}_{y_1}^{\mathsf{lb}}(\boldsymbol{x}; f_{\mathrm{b}}, \phi_\psi, \alpha)$
2: $t_2^{\mathsf{ub}} \leftarrow \max_{y \neq y_1} \hat{p}_y^{\mathsf{ub}}(\boldsymbol{x}; f_{\mathrm{b}}, \phi_\psi, \alpha)$
3: **for** $r = 0$ **to** $\infty$ **do**
4:     **for all** $(n_{\mathsf{del}}, n_{\mathsf{ins}}, n_{\mathsf{sub}}) \in \mathcal{C}(\mathsf{o}, r)$ **do**
5:         $|\bar{\boldsymbol{x}}| \leftarrow |\boldsymbol{x}| + n_{\mathsf{ins}} - n_{\mathsf{del}}$
6:         $\bar{t}_1^{\mathsf{lb}} \leftarrow \mathsf{lb}\big(t_1^{\mathsf{lb}}, \boldsymbol{x}, \bar{\boldsymbol{x}}, \psi\big)$
7:         $\bar{t}_2^{\mathsf{ub}} \leftarrow \mathsf{ub}\big(t_2^{\mathsf{ub}}, \boldsymbol{x}, \bar{\boldsymbol{x}}, \psi\big)$
8:         **if** $\bar{t}_1^{\mathsf{lb}} \leq \bar{t}_2^{\mathsf{ub}}$ **then**
9:             **return** $r$

---

top and runner-up using an independent set of samples and output radius of $0$ if there is disagreement between the two batches of samples. The for loop in Line 3 then attempts to verify a certificate at increasing radii, by checking all neighboring inputs parameterized by the number of edits of each type (Line 4). If a violation is found—where the classifier's prediction changes due to the runner-up probability exceeding the probability for $y_1$ within the $\alpha$ confidence bound—the previous best radius is returned. The proof of Theorem 4 is deferred to Appendix A.

**Theorem 4.** *Given a smoothed classifier $f$ as described in Algorithm 2 with a length-dependent deletion function $\psi$,*

$$\forall \bar{\boldsymbol{x}} \in B_r(\boldsymbol{x}; \mathsf{o}) : f(\boldsymbol{x}) = f(\bar{\boldsymbol{x}})$$

*with confidence level at least $1 - \alpha$.*

We now consider various options for specifying a length-dependent deletion function $\psi$.

**Fixed-Rate Deletion [17]** The most basic option is a fixed-rate deletion mechanism,

$$\psi(\boldsymbol{x}) \coloneqq p_{\mathsf{del}}, \tag{4}$$

for some constant $p_{\mathsf{del}} \in [0, 1]$. For this specialization, we show in Appendix A that the Lemma 3 simplifies approximately to the closed form analytical solutions obtained by Huang et al. [18]. Although this mechanism was shown to be effective for malware detection and some natural language tasks, it is less suited when there is variation in sequence/text lengths. In particular, one can optimize the mechanism by setting $p_{\mathsf{del}}$ for best average length input performance, but this may be suboptimal for inputs that deviate from the average length.

**Length Dependent Deletion** We propose a mechanism, AdaptDel, where the deletion rate $\psi$ smoothly increases as a function of the input sequence length. Specifically, we define

$$\psi(\boldsymbol{x}) \coloneqq \max \left( p_{\mathsf{lb}}, p \left( 1 - \frac{k}{|\boldsymbol{x}|} \right) \right), \tag{5}$$

where $p_{\text{lb}} \in [0, 1]$ is a minimum deletion rate, $p \in [p_{\text{lb}}, 1]$ is the asymptotic deletion rate (as $|\boldsymbol{x}| \to \infty$), and $k > 0$ is a parameter that scales the deletion rate based on the input length. In subsequent experiments, we set $p_{\text{lb}} = p_{\text{del}}$, $p = 1$ and $k = \lfloor (1 - p_{\text{lb}}) \mathbb{E}\{|\boldsymbol{x}|\} \rfloor$ so that the deletion rate in Equation 5 matches the deletion rate in Equation 4 for a test sequence of average length. This allows for a fair comparison between fixed-rate deletion and AdaptDel. We note that clipping of the deletion rate below $p_{\text{lb}}$ is done to avoid a certified radius of zero for short inputs. While this may sacrifice some accuracy, it enables non-vacuous certificates for shorter sequences. One intuition is that longer input sequences are inherently more tolerant to noise, allowing for a more significant deletion rate to be applied without severely impacting model performance.

**Optimized Length Dependent Deletion**  We also consider an enhanced optimization based certification approach, dubbed AdaptDel+, where we bin the inputs by sizes into $n$ bins and find an optimal expected length after deletion for each bin using golden section based search (Algorithm 5). In particular, let $g(\boldsymbol{x})$ denote the index of the bin input $\boldsymbol{x}$ belongs to, then we have $\psi(\boldsymbol{x}) \coloneqq 1 - \frac{k_{g(\boldsymbol{x})}}{|\boldsymbol{x}|}$ where $k_{g(\boldsymbol{x})}$ is the empirically optimized expected length for bin $g(\boldsymbol{x})$. We provide more details and intuition for Algorithm 5 in the Appendix C.

## 4  Experiments

To evaluate the effectiveness of AdaptDel and AdaptDel+, we conduct experiments on a diverse set of natural language processing tasks. These include five-class sentiment analysis on the Yelp dataset [22], spam detection using the SpamAssassin dataset [30, 31], sentiment analysis on the IMDB dataset [32], and unreliable news detection using the LUN dataset [33, 31]. The varying input sizes of these datasets allow for a comprehensive assessment of AdaptDel's overall effectiveness. We also analyze performance by dividing inputs into quartiles based on their length. Detailed specifications of these datasets are provided in Table 1 in Appendix D. The appendices provide further certified results (Appendix E), details on computational efficiency (Appendix F) and a supplementary analysis of empirical robustness against common text attacks (Appendix G),

**Model and Parameter Setup**  We use a pre-trained RoBERTa model [34] as a non-smoothed baseline (NoSmooth) and as a base model for randomized smoothing. We compare our adaptive deletion rate policies AdaptDel and AdaptDel+ with a fixed deletion rate baseline, that corresponds to CERT-ED [18]. All of these methods support certification of edit distance. We note that CERT-ED has one parameter $p_{\text{del}}$, AdaptDel has two parameters $p_{\text{lb}}$ and $k$. The parameters for AdaptDel+ are determined via an empirical calibration procedure on the training set (see Appendix C). We additionally include RanMASK [13] as a baseline that supports a more limited Hamming distance certificate. It uses a masking mechanism, where the fraction of masked tokens $p_{\text{mask}}$ is fixed. For all smoothed models, we apply the same base RoBERTa model, training procedure, and parameter settings where possible. During inference, we use 1000 samples for prediction and 4000 samples for certification to ensure stable and reliable results. We refer the reader to Appendix D for further details, including how AdaptDel+ is calibrated.

**Performance Measures**  We report *clean accuracy*, the accuracy without any robustness guarantee, as a measure of model performance. To evaluate robustness, we use the *certified radius* (CR), which quantifies the largest perturbation radius under which a prediction is provably robust. A larger CR indicates greater robustness. Following Huang et al. [18], we also report *log-cardinality of the certified region* $(\log(\text{CC}))$, the base-10 logarithm of the number of perturbations enclosed by the CR under the given threat model. For Hamming distance (RanMASK), this is an exact count, whereas for edit distance (CERT-ED, AdaptDel, AdaptDel+), it serves as a lower bound, underestimating by at most one order of magnitude. For incorrectly classified inputs, we assign $\text{CR} = \text{CC} = 0$. Finally, we define *certified accuracy* as the proportion of inputs that are both correctly classified and possess log-cardinalities exceeding a given threshold $c$.

### 4.1  Robustness-Accuracy Trade-off

Figure 1 presents the certified accuracy as a function of the log-cardinality of the certified region for all datasets. Each curve depicts how the fraction of test instances that remain correct and certifiably robust changes as the size of the certificate grows (in log-scale). Combined with results in

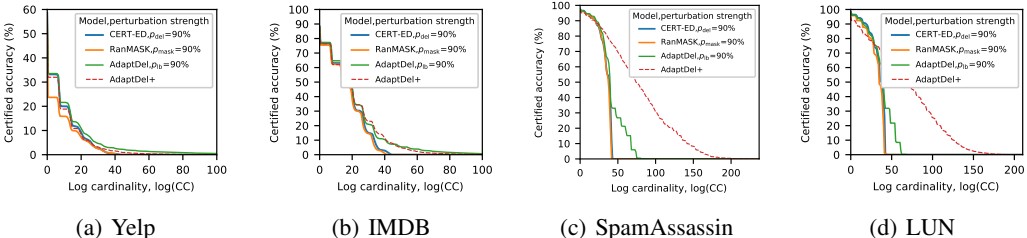

|     |     |     |     |
| --- | --- | --- | --- |
| (a) Yelp | (b) IMDB | (c) SpamAssassin | (d) LUN |

Figure 1: Certified accuracy plotted against a lower bound on the log-cardinality of the certified region. Each point $(c, a)$ on a curve indicates that a fraction $a$ of the test inputs were correctly classified with a certified log-cardinality of at least $c$. While AdaptDel consistently outperforms the baselines, AdaptDel+ achieves the highest certified accuracy for larger certified regions.

Table 4 in Appendix E.2, both AdaptDel and AdaptDel+ maintain competitive clean accuracy (0–2% drop compared to RanMASKand CERT-ED), while significantly enhancing robustness guarantees compared to RanMASK and CERT-ED. In particular, they improve the mean certified radius by an average of over 50%, and the median cardinality of the certified region by up to *30 orders of magnitude*. This substantial expansion highlights the effectiveness of adaptive deletion in certifying robustness against a significantly broader space of perturbations, making it particularly valuable in adversarial settings where sequence manipulations are common.

We observe that methods based on AdaptDel and AdaptDel+ generally yield higher certified accuracy than CERT-ED and RanMASK for larger cardinalities, reflecting the same trend of increased robustness reported in our mean CR and median CC metrics. These improvements are especially pronounced for SpamAssassin and LUN datasets which contain more uniform length variations (see quantile of lengths in Figure 2 and Figure 4). Our results show improved performance for longer sequences that deviate from the average length.

## 4.2 Robustness Scaling with Input Size

Figure 2 shows how certified accuracy scales with respect to both input size (grouped by quartile) and the log-cardinality of the certified region. Each panel/facet on the left of the figure corresponds to a quartile, where the range of input sizes can be read from the quantile function on the right. For the Yelp dataset (left), results within the first two quarters are slightly mixed with AdaptDel and AdaptDel+ showing moderate advantage over RanMASK and CERT-ED. This is expected, as AdaptDel is configured to match the fixed deletion rate of CERT-ED for short sequences, with its advantages becoming more pronounced as sequence length increases. However, for the fourth quarter where the length variations is concentrated, both AdaptDel and AdaptDel+ achieve substantial gains in certified accuracy, indicating that as input size grows, our smoothing techniques can certify robustness against a broader range of adversarial edits. Meanwhile, for the SpamAssassin dataset (right), a similar pattern emerges, but the benefits are more pronounced. Additionally, we observe that AdaptDel+ maintains significantly stronger robustness guarantees across all lengths, showing the effectiveness of our calibration strategy in enhancing certified accuracy for varying input sizes.

Overall, our approach not only preserves clean accuracy but significantly enhances certified robustness, particularly for longer sequences. Compared to state-of-the-art methods, we achieve an average improvement in mean certified radius of over 50% and increase the median certified region cardinality by up to 30 orders of magnitude. This demonstrates superior scalability and highlights the practical value of tailoring the smoothing rate to input length, a property that can be critical in adversarial settings where length variations are common.

## 5 Related Work

In addition to the works mentioned in the introduction, we now describe approaches closest to ours.

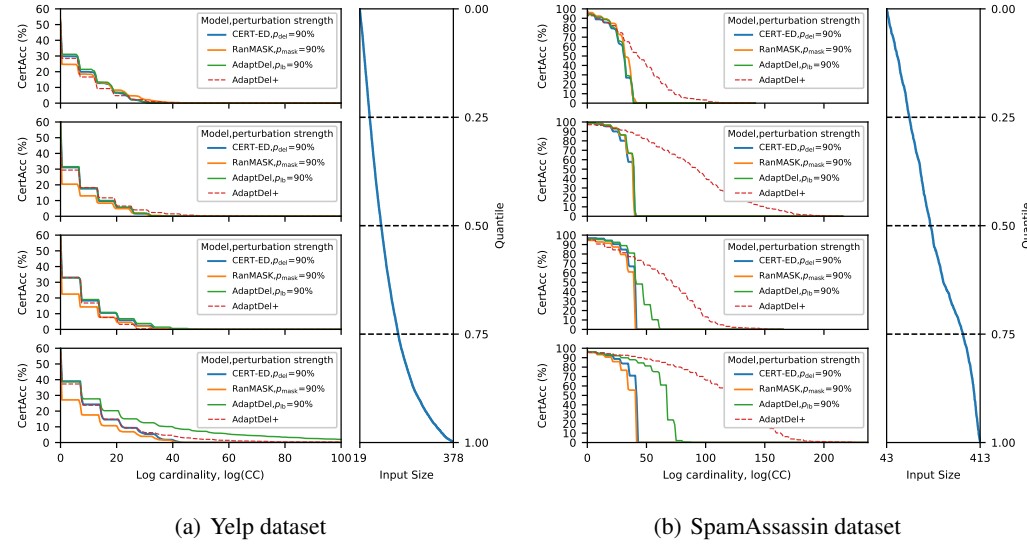

(a) Yelp dataset          (b) SpamAssassin dataset

Figure 2: Certified accuracy as a function of the log-cardinality of the certified region, grouped by quartile of input size. The subfigure on the right displays the quantile by input size, with the dashed lines indicating the quartiles corresponding to the certified accuracy plots on the left. Each set of axes (top to bottom) corresponds to a split of the test set on the length-based quartiles (smallest to largest). For example, the second plot from top to bottom shows the certified accuracies of examples within Q1 to Q2. The results demonstrate that the methods scale effectively across varying input sizes, with higher certified accuracy achieved for larger input sizes and higher log-cardinality regions.

**Certification for Edit Perturbations**      There is a growing body of work aimed at extending certification to handle edit-based threat models on structured input spaces. In the natural language domain, randomized smoothing has been applied to certify robustness against synonym and word substitution attacks [11–13, 35]. However, their certificates do not cover insertions or deletions of words. To address this gap, recent work has studied broader threat models that go beyond substitutions. Zhang et al. [16] provide certificates against permutations and $\ell_2$-bounded perturbations of sequence elements in embedding space. Rocamora et al. [36] provide edit distance certificates for convolutional classifiers with bounded Lipschitz constants. While their approach does not require smoothing, it is limited to convolutional architectures and yields empirical accuracy and robustness guarantees that are significantly smaller than those of the randomized smoothing approach considered in this paper.

Beyond sequences, edit distance certification has also been explored for sets and graphs. Liu et al. [37] apply subsampling-based randomized smoothing to certify point cloud classifiers against a bounded number of point insertions, deletions, and substitutions. Schuchardt et al. [38] propose a general framework for group equivariant tasks that provides edit distance certificates for graph classifiers [39].

**Input-Dependent Smoothing**      Most applications of randomized smoothing consider additive noise mechanisms, where the noise level is independent of the input. While this simplifies the robustness analysis, it is suboptimal and has been shown to result in disparities in class-wise accuracy [40]. Several works have attempted to address this, particularly for input-dependent Gaussian noise. Wang et al. [5] divide the input space into several "robust regions", each with a constant calibrated noise-level, and use the fixed-noise certificate of Cohen et al. [14] within each region. However, their approach assumes all test samples are available for calibration and it comes with higher train/test costs. Eiras et al. [19] and Alfarra et al. [20] also use the fixed-noise certificate as a starting point, but before issuing a new certificate they apply a correction that depends on all previously issued certificates. By contrast, Súkeník et al. [21] obtain a tight certificate for input-dependent Gaussian noise by generalizing the proof technique of Cohen et al. [14]. They find that the variability of the noise scale is limited in practice due to the curse of dimensionality. Lyu et al. [41] propose a general

framework for input-dependent smoothing mechanisms that can potentially consist of multiple steps. However, in order to obtain a certificate, the mechanism must satisfy functional differential privacy.

While our approach shares some similarities with the above prior work, it differs fundamentally in two ways. First, the edit distance certificates we consider include inputs with variable dimensions (sequence lengths), which is not the case for $\ell_p$ certificates where all inputs have the same dimensions. This makes deriving a sound certificate for dimension-dependent noise more challenging in our case. We note that Gaussian smoothing with a dimension-dependent noise scale is covered by the standard fixed-noise certificate of Cohen et al. [14]. Cohen et al. observed that higher dimensional images can tolerate large noise scales while retaining visual information—we enable such noise scaling for variable-length sequences while ensuring sound certification. Second, our certificate and mechanism are qualitatively very different—our deletion mechanism cannot be expressed as additive noise, and edit distance is arguably more challenging to analyze than $\ell_p$ distance given it is defined as the solution to a discrete optimization problem.

## 6    Conclusion

In this work we propose variable rate deletion for sequence classification—extending randomized smoothing certification to adapt to inputs of varying length. We develop a theoretical framework for variable rate deletion, which allows the deletion probability to adapt dynamically based on input properties. Building on this foundation, we propose AdaptDel and AdaptDel+, mechanisms designed to enhance robustness certification while maintaining competitive performance. The former uses a length-dependent deletion rate, whereas the latter further optimizes the deletion rate using input binning and empirical calibration.

Our results demonstrate that AdaptDel and AdaptDel+ consistently outperform existing state-of-the-art methods, such as CERT-ED and RanMASK, across diverse NLP tasks with varying input lengths. Our methods achieve a superior trade-off, offering significant gains in certified robustness while maintaining competitive clean accuracy. This highlights their effectiveness in real-world scenarios (*e.g.* LLMs) where input sequences can vary significantly in length. Our contributions pave the way for more adaptable and robust approaches to certifying sequence classifiers.

## Limitations

Our work focuses on the problem of certified robustness for sequence classification against edit distance perturbations. While edit distance offers a more appropriate threat model than $\ell_p$-bounded attacks, this choice is still a syntactic surrogate for attacks that preserve semantic similarity. And while our contributions permit length-dependent deletion rates, it is conceivable that more general input-dependent deletion rates could yield improved certified robustness.

## Acknowledgments and Disclosure of Funding

We acknowledge partial funding from the Australian Research Council DP220102269. This research was supported by The University of Melbourne's Research Computing Services and the Petascale Campus Initiative.

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

# A  Proofs and Additional Results

This appendix provides proofs for the results in Sections 3.3 and 3.4. For the proof of Lemma 2, we refer the reader Lemma 4 of Huang et al. [17]. We also provide an additional result (Lemma 7) showing that our certificate for variable rate deletion recovers the certificate of Huang et al. [17] for fixed rate deletion, with only a negligible difference.

Before proving Lemma 3, we obtain bounds on the sums that appear in the decomposition of the smoothed class probability $p_y(\bar{\boldsymbol{x}}; f_{\mathrm{b}})$ presented in Equation 3.

**Lemma 5.**

$$\sum_{\boldsymbol{\epsilon} \in \mathcal{E}(\boldsymbol{x}) \backslash \mathcal{E}(\boldsymbol{x}, \boldsymbol{\epsilon}^\star)} s_{f_{\mathrm{b}}}(\boldsymbol{\epsilon}, \boldsymbol{x}) \leq 1 - \psi^{|\boldsymbol{x}| - |\boldsymbol{z}^\star|} \tag{6}$$

$$\sum_{\boldsymbol{\epsilon} \in \mathcal{E}(\boldsymbol{x}, \boldsymbol{\epsilon}^\star)} s_{f_{\mathrm{b}}}(\boldsymbol{\epsilon}, \boldsymbol{x}) \geq p_y(\boldsymbol{x}; f_{\mathrm{b}}) - 1 + \psi^{|\boldsymbol{x}| - |\boldsymbol{z}^\star|}. \tag{7}$$

*Proof.* First, observe that

$$\sum_{\boldsymbol{\epsilon} \in \mathcal{E}(\boldsymbol{x}) \backslash \mathcal{E}(\boldsymbol{x}, \boldsymbol{\epsilon}^\star)} s_{f_{\mathrm{b}}}(\boldsymbol{\epsilon}, \boldsymbol{x}) = 1 - \sum_{\boldsymbol{\epsilon} \in \mathcal{E}(\boldsymbol{x}, \boldsymbol{\epsilon}^\star)} s_{f_{\mathrm{b}}}(\boldsymbol{\epsilon}, \boldsymbol{x})$$

$$\leq 1 - \sum_{\boldsymbol{\epsilon} \in \mathcal{E}(\boldsymbol{x}, \boldsymbol{\epsilon}^\star)} q(\boldsymbol{\epsilon}|\boldsymbol{x})$$

$$= 1 - \psi^{|\boldsymbol{x}| - |\boldsymbol{z}^\star|} \sum_{\boldsymbol{\epsilon} \in \mathcal{E}(\boldsymbol{x}, \boldsymbol{\epsilon}^\star)} q(\boldsymbol{\epsilon}^\star - \boldsymbol{\epsilon}|\boldsymbol{x})$$

$$\leq 1 - \psi^{|\boldsymbol{x}| - |\boldsymbol{z}^\star|}.$$

Then it is straightforward to decompose the smoothed class probability as

$$p_y(\boldsymbol{x}; f_{\mathrm{b}}) = \sum_{\boldsymbol{\epsilon} \in \mathcal{E}(\boldsymbol{x}, \boldsymbol{\epsilon}^\star)} s_{f_{\mathrm{b}}}(\boldsymbol{\epsilon}, \boldsymbol{x}) + \sum_{\boldsymbol{\epsilon} \in \mathcal{E}(\boldsymbol{x}) \backslash \mathcal{E}(\boldsymbol{x}, \boldsymbol{\epsilon}^\star)} s_{f_{\mathrm{b}}}(\boldsymbol{\epsilon}, \boldsymbol{x})$$

and obtain the lower bound

$$\sum_{\boldsymbol{\epsilon} \in \mathcal{E}(\boldsymbol{x}, \boldsymbol{\epsilon}^\star)} s_{f_{\mathrm{b}}}(\boldsymbol{\epsilon}, \boldsymbol{x}) = p_y(\boldsymbol{x}; f_{\mathrm{b}}) - \sum_{\boldsymbol{\epsilon} \in \mathcal{E}(\boldsymbol{x}) \backslash \mathcal{E}(\boldsymbol{x}, \boldsymbol{\epsilon}^\star)} s_{f_{\mathrm{b}}}(\boldsymbol{\epsilon}, \boldsymbol{x})$$

$$\geq p_y(\boldsymbol{x}; f_{\mathrm{b}}) - 1 + \psi^{|\boldsymbol{x}| - |\boldsymbol{z}^\star|}.$$

$\square$

Our proof strategy for Lemma 3, restated below, is inspired by the analysis of Lee et al. [42] for continuous-space smoothing. However, several key adaptations are necessary for the discrete, variable-length setting. First, their analysis relies on partitioning the space of perturbed inputs and ordering the parts by increasing likelihood. This is not possible in our case as this likelihood is not tractable due to the fact that multiple paths can lead to the same perturbed input. Second, discrete space requires careful treatment of rounding to ensure validity of the bound.

**Lemma 3.** *Let $\boldsymbol{x}, \bar{\boldsymbol{x}} \in \mathcal{X}$ be a pair of inputs with a longest common subsequence (LCS) $\boldsymbol{z}^\star$ and let $\mu = p_y(\boldsymbol{x}; f_{\mathrm{b}})$. Define*

$$H^* = \begin{cases} \min_{h : \sum_{i=0}^{h} \mathcal{B}_i(|\boldsymbol{z}^\star|, \psi) \geq \mu - 1 + \psi^{|\boldsymbol{x}| - |\boldsymbol{z}^\star|}} h, & \psi \geq \bar{\psi}, \\ \max_{h : \sum_{i=h}^{|\boldsymbol{z}^\star|} \mathcal{B}_i(|\boldsymbol{z}^\star|, \psi) \geq \mu - 1 + \psi^{|\boldsymbol{x}| - |\boldsymbol{z}^\star|}} h, & \psi < \bar{\psi}, \end{cases}$$

*as a threshold on the number of tokens retained when editing $\boldsymbol{x}$, where $\mathcal{B}_k(n, p) := \binom{n}{k}(1 - p)^k p^{n-k}$ is the Binomial pmf for $n$ trials with success probability $1 - p$. Then there exists a lower bound*

$\text{lb}(\mu, \boldsymbol{x}, \bar{\boldsymbol{x}}, \psi) \le p_y(\bar{\boldsymbol{x}}; f_{\mathrm{b}})$ *such that:*

$$\text{lb}(\mu, \boldsymbol{x}, \bar{\boldsymbol{x}}, \psi) = \frac{\bar{\psi}^{|\bar{\boldsymbol{x}}| - |\boldsymbol{z}^\star|}}{\psi^{|\boldsymbol{x}| - |\boldsymbol{z}^\star|}} \left( \sum_{i=\boldsymbol{l}(H^*+1)}^{(1-\boldsymbol{l})(H^*-1)+\boldsymbol{l}|\boldsymbol{z}^\star|} \mathcal{B}_i(|\boldsymbol{z}^\star|, \bar{\psi}) \right.$$

$$\left. + \mathcal{B}_{H^*}(|\boldsymbol{z}^\star|, \bar{\psi}) \left\lfloor \frac{c(\mu, |\boldsymbol{x}|, |\boldsymbol{z}^\star|, \psi, H^*)}{\mathcal{B}_{H^*}(|\boldsymbol{z}^\star|, \psi)} \right\rfloor_{\binom{|\boldsymbol{z}^\star|}{H^*}^{-1}} \right),$$

*where*

$$c(\mu, |\boldsymbol{x}|, |\boldsymbol{z}^\star|, \psi, H^*) = \mu - 1 + \psi^{|\boldsymbol{x}| - |\boldsymbol{z}^\star|} - \sum_{i=\boldsymbol{l}(H^*+1)}^{(1-\boldsymbol{l})(H^*-1)+\boldsymbol{l}|\boldsymbol{z}^\star|} \mathcal{B}_j(|\boldsymbol{z}^\star|, \psi),$$

$\boldsymbol{l} = \mathbf{1}_{\psi < \bar{\psi}}$ *is a binary indicator and* $\lfloor \cdot \rfloor_v := \lfloor \frac{\cdot}{v} \rfloor v$ *is a gridded flooring operation.*

*Proof.* We will prove the case where $\psi \ge \bar{\psi}$, the proof for the opposite case follows a similar logic. By dropping the second sum in Equation 3, we obtain the following lower bound:

$$p_y(\bar{\boldsymbol{x}}; f_{\mathrm{b}}) \ge \frac{\bar{\psi}^{|\bar{\boldsymbol{x}}|}}{\psi^{|\boldsymbol{x}|}} \underbrace{\sum_{\boldsymbol{\epsilon} \in \mathcal{E}(\boldsymbol{x}, \boldsymbol{\epsilon}^\star)} \rho(\psi, \bar{\psi})^{|\boldsymbol{\epsilon}|} s_{f_{\mathrm{b}}}(\boldsymbol{\epsilon}, \boldsymbol{x})}_{\sigma(\bar{\boldsymbol{x}}; f_{\mathrm{b}})}.$$

Since $\frac{\bar{\psi}^{|\bar{\boldsymbol{x}}|}}{\psi^{|\boldsymbol{x}|}}$ is a non-negative constant, we focus on lower bounding the summation term $\sigma(\bar{\boldsymbol{x}}; f_{\mathrm{b}})$ and reinstate the constant later.

Let $\mathcal{F} = \{\bar{f} \mid p_y(\boldsymbol{x}; \bar{f}) = \mu\}$ be the set of base classifiers that are consistent with the observed value of the smoothed class probability $\mu = p_y(\boldsymbol{x}; f_{\mathrm{b}})$. We seek a lower bound on $\sigma(\bar{\boldsymbol{x}}, f_{\mathrm{b}})$ that does not depend on the functional form of the base classifier, so we consider the worst case in the set $\mathcal{F}$:

$$\sigma(\bar{\boldsymbol{x}}; f_{\mathrm{b}}) \ge \min_{\bar{f} \in \mathcal{F}} \sum_{\boldsymbol{\epsilon} \in \mathcal{E}(\boldsymbol{x}, \boldsymbol{\epsilon}^\star)} \rho(\psi, \bar{\psi})^{|\boldsymbol{\epsilon}|} s_{\bar{f}}(\boldsymbol{\epsilon}, \boldsymbol{x}). \tag{8}$$

We can rewrite the sum over $\mathcal{E}(\boldsymbol{x}, \boldsymbol{\epsilon}^\star)$ as a double sum over the number of retained elements $i$, followed by a sum over edits in the set $\mathcal{E}_i(\boldsymbol{x}, \boldsymbol{\epsilon}^\star) = \{\boldsymbol{\epsilon} \in \mathcal{E}(\boldsymbol{x}, \boldsymbol{\epsilon}^\star) \mid |\boldsymbol{\epsilon}| = i\}$. Hence we have

$$\text{RHS of (8)} = \min_{\bar{f} \in \mathcal{F}} \sum_{i=0}^{|\boldsymbol{z}^\star|} \rho(\psi, \bar{\psi})^i \sum_{\boldsymbol{\epsilon} \in \mathcal{E}_i(\boldsymbol{x}, \boldsymbol{\epsilon}^\star)} s_{\bar{f}}(\boldsymbol{\epsilon}, \boldsymbol{x})$$

$$= \min_{\bar{f} \in \mathcal{F}} \sum_{i=0}^{|\boldsymbol{z}^\star|} \psi^{|\boldsymbol{z}^\star|} \bar{\psi}^{-i}(1 - \bar{\psi})^i \sum_{\boldsymbol{\epsilon} \in \mathcal{E}_i(\boldsymbol{x}, \boldsymbol{\epsilon}^\star)} \mathbf{1}_{\bar{f}(\text{apply}(\boldsymbol{x}, \boldsymbol{\epsilon})) = y}, \tag{9}$$

where the second equality follows from the definitions of $s_{\bar{f}}(\boldsymbol{\epsilon}, \boldsymbol{x})$ and $\rho(\psi, \bar{\psi})$ in Lemma 2 and the i.i.d. Bernoulli specification of $q(\boldsymbol{\epsilon}|\boldsymbol{x})$.

Let $[n] = \{0, \ldots, n\}$ denote the set of non-negative integers up to $n$. We can rewrite the minimization problem in Equation 9 as a minimization problem over the set of functions $\mathcal{H}$ from $[N]$ to $[M]$ where $N = |\boldsymbol{z}^\star|$ and $M = \bigcup_{i \in [N]} \binom{N}{i}$. Specifically, we have

$$\min_{h \in \mathcal{H}} \quad \sum_{i=0}^{N} v(i) h(i)$$

$$\text{s.t.} \quad h(i) \equiv \sum_{\boldsymbol{\epsilon} \in \mathcal{E}_i(\boldsymbol{x}, \boldsymbol{\epsilon}^\star)} \mathbf{1}_{\bar{f}(\text{apply}(\boldsymbol{x}, \boldsymbol{\epsilon})) = y} \quad \text{and} \quad \bar{f} \in \mathcal{F}, \tag{10}$$

where we have defined $v(i) := \psi^N \bar{\psi}^{-i}(1 - \bar{\psi})^i$. Next, we continue to lower bound the solution to the above problem by relaxing the constraints. First, we replace $\mathcal{F}$ by a superset (invoking Lemma 5):

$$\mathcal{F}' = \left\{ \bar{f} \mid \sum_{i=0}^{N} w(i) \sum_{\boldsymbol{\epsilon} \in \mathcal{E}_i(\boldsymbol{x}, \boldsymbol{\epsilon}^\star)} \mathbf{1}_{\bar{f}(\text{apply}(\boldsymbol{x}, \boldsymbol{\epsilon})) = y} \ge W \right\},$$

where we have defined $w(i) := (1 - \psi)^i \psi^{N-i}$ and $W := \mu - 1 + \psi^{|\boldsymbol{x}|-N}$. The new constraint on $h$ (replacing Equation 10) is then

$$h(i) \equiv \sum_{\boldsymbol{\epsilon} \in \mathcal{E}_i(\boldsymbol{x}, \boldsymbol{\epsilon}^\star)} \mathbf{1}_{\bar{f}(\text{apply}(\boldsymbol{x}, \boldsymbol{\epsilon}))=y} \quad \text{and} \quad \sum_{i=0}^{N} w(i)h(i) \geq W \quad \text{and} \quad \bar{f} \in \mathcal{X} \rightarrow \mathcal{Y}.$$

Second, we relax the constraint further, by allowing $h(i)$ to be any integer in the set $[\binom{N}{i}]$.

The resulting relaxed minimization problem is a bounded knapsack problem:

$$\min_{h \in \mathcal{H}} \quad \sum_{i=0}^{N} v(i)h(i) \tag{11}$$

$$\text{s.t.} \quad \sum_{i=0}^{N} w(i)h(i) \geq W \quad \text{and} \quad h(i) \leq \binom{N}{i}. \tag{12}$$

Noting that the value per unit weight $v(i)/w(i) = \rho(\psi, \bar{\psi})^i$ is increasing in $i$ for $\psi \geq \bar{\psi}$, we propose the following greedy "solution", which slightly violates the constraint:

$$h^*(i) = \begin{cases} \binom{N}{i}, & i < H^* \\ \left\lfloor \frac{W - \sum_{j=0}^{H^*-1} \binom{N}{i} w(j)}{w(H^*)} \right\rfloor, & i = H^* \\ 0, & \text{otherwise}, \end{cases}$$

where $H^* = \min_{H:\sum_{i=0}^{H} \binom{N}{i} w(j) \geq W} H$.

We now show that $h^*$ yields a valid lower bound on Equation 11. First, observe that

$$\sum_{i=0}^{N} w(i)h^*(i) \leq W, \tag{13}$$

which means the constraint in Equation 12 is not necessarily satisfied by $h^*$. Now suppose there exists a solution $h'$ to Equation 11 that yields a tighter lower bound than $h^*$. Define $\Delta(i) := w(i)(h^*(i) - h'(i))$ and observe that $\Delta(i) \geq 0$ when $i < H^*$ and $\Delta(i) \leq 0$ when $i > H^*$. By combining Equation 13 (for $h^*$) and Equation 12 (for $h'$), we have $\sum_{i=0}^{N} \Delta(i) \leq 0$. Together, these results imply

$$\sum_{i=0}^{N} \rho(\psi, \bar{\psi})^i \Delta(i) \leq \rho(\psi, \bar{\psi})^{H^*} \sum_{i=0}^{N} \Delta(i) \leq 0.$$

The first inequality holds because we are increasing the magnitude of positive terms ($i < H^*$) and decreasing the magnitude of negative terms ($i > H^*$), noting that $\rho(\psi, \bar{\psi})^i$ is increasing in $i$. This then implies (by expanding the definition of $\Delta(i)$):

$$\sum_{i=0}^{N} v(i)h^*(i) = \sum_{i=0}^{N} \rho(\psi, \bar{\psi})^i w(i)h^*(i) \leq \sum_{i=0}^{N} \rho(\psi, \bar{\psi})^i w(i)h'(i) = \sum_{i=0}^{N} v(i)h'(i),$$

which is a contradiction, thereby confirming that $h^*$ yields a valid lower bound.

Substituting $h^*$ in the objective yields the following lower bound:

$$\sigma(\bar{\boldsymbol{x}}; f_\text{b}) \geq \sum_{i=0}^{N} v(i)h^*(i)$$

$$= \left( \frac{\psi}{\bar{\psi}} \right)^{|\boldsymbol{z}^\star|} \left\{ \sum_{i=0}^{H^*-1} \mathcal{B}_i(|\boldsymbol{z}^\star|, \bar{\psi}) \right.$$

$$\left. + \mathcal{B}_{H^*}(|\boldsymbol{z}^\star|, \bar{\psi}) \left\lfloor \frac{\mu - 1 + \psi^{|\boldsymbol{x}|-|\boldsymbol{z}^\star|} - \sum_{j=0}^{H^*-1} \mathcal{B}_j(|\boldsymbol{z}^\star|, \psi)}{\mathcal{B}_{H^*}(|\boldsymbol{z}^\star|, \psi)} \right\rfloor_{\binom{|\boldsymbol{z}^\star|}{H^*}^{-1}} \right\}.$$

Finally, we multiple both sides of the above inequality by $\frac{\bar{\psi}^{|\bar{\boldsymbol{x}}|}}{\psi^{|\boldsymbol{x}|}}$ to obtain the desired lower bound on $p_y(\bar{\boldsymbol{x}}; f_{\mathrm{b}})$.

$\square$

We use the same proof strategy to obtain an upper bound on the smoothed probability below. The upper bound can be used to derive a robustness certificate for in the style of Lecuyer et al. [15].

**Lemma 6.** *Let* $\boldsymbol{x}, \bar{\boldsymbol{x}} \in \mathcal{X}$ *be a pair of inputs with a longest common subsequence (LCS)* $\boldsymbol{z}^\star$ *and* $\mu = p_y(\boldsymbol{x}; f_{\mathrm{b}})$. *Define*

$$H^* = \begin{cases} \min_{H:\sum_{i=0}^{H} \mathcal{B}_i(|\boldsymbol{z}^\star|,\psi) \geq \mu} H, & \psi \leq \bar{\psi}, \\ \max_{H:\sum_{i=H}^{|\boldsymbol{z}^\star|} \mathcal{B}_i(|\boldsymbol{z}^\star|,\psi) \geq \mu} H, & \psi > \bar{\psi}, \end{cases}$$

*as a threshold on the number of tokens retained when editing* $\boldsymbol{x}$, *where* $\mathcal{B}_k(n,p) := \binom{n}{k}(1-p)^k p^{n-k}$ *is the Binomial pmf for* $n$ *trials with with success probability* $1 - p$. *Then there exists an upper bound* $\mathrm{ub}(\mu, \boldsymbol{x}, \bar{\boldsymbol{x}}, \psi) \geq p_y(\bar{\boldsymbol{x}}; f_{\mathrm{b}})$ *such that:*

$$\mathrm{ub}(\mu, \boldsymbol{x}, \bar{\boldsymbol{x}}, \psi) = \frac{\bar{\psi}^{|\bar{\boldsymbol{x}}|-|\boldsymbol{z}^\star|}}{\psi^{|\boldsymbol{x}|-|\boldsymbol{z}^\star|}} \left( \sum_{i=\boldsymbol{g}(H^*+1)}^{(1-\boldsymbol{g})(H^*-1)+\boldsymbol{g}|\boldsymbol{z}^\star|} \mathcal{B}_i(|\boldsymbol{z}^\star|, \bar{\psi}) \right.$$

$$\left. + \mathcal{B}_{H^*}(|\boldsymbol{z}^\star|, \bar{\psi}) \left\lceil \frac{c(\mu, |\boldsymbol{z}^\star|, \psi, H^*)}{\mathcal{B}_{H^*}(|\boldsymbol{z}^\star|, \psi)} \right\rceil_{\binom{|\boldsymbol{z}^\star|}{H^*}^{-1}} \right) + 1 - \bar{\psi}^{|\bar{\boldsymbol{x}}|-|\boldsymbol{z}^\star|},$$

*where*

$$c(\mu, |\boldsymbol{z}^\star|, \psi, H^*) = \mu - \sum_{i=\boldsymbol{g}(H^*+1)}^{(1-\boldsymbol{g})(H^*-1)+\boldsymbol{g}|\boldsymbol{z}^\star|} \mathcal{B}_j(|\boldsymbol{z}^\star|, \psi),$$

$\boldsymbol{g} = \mathbf{1}_{\psi > \bar{\psi}}$ *is a binary indicator and* $\lceil \cdot \rceil_v := \lceil \frac{\cdot}{v} \rceil v$ *is a gridded ceiling operation.*

*Proof.* We will prove the case where $\psi \leq \bar{\psi}$. By combining Equations 3 and 6, we obtain the following upper bound:

$$p_y(\bar{\boldsymbol{x}}; f_{\mathrm{b}}) \leq \frac{\bar{\psi}^{|\bar{\boldsymbol{x}}|}}{\psi^{|\boldsymbol{x}|}} \underbrace{\sum_{\boldsymbol{\epsilon} \in \mathcal{E}(\boldsymbol{x}, \boldsymbol{\epsilon}^\star)} \rho(\psi, \bar{\psi})^{|\boldsymbol{\epsilon}|} s_{f_{\mathrm{b}}}(\boldsymbol{\epsilon}, \boldsymbol{x})}_{\sigma(\bar{\boldsymbol{x}}; f_{\mathrm{b}})} + \left( 1 - \bar{\psi}^{|\bar{\boldsymbol{x}}|-|\boldsymbol{z}^\star|} \right).$$

We focus on upper bounding the summation term $\sigma(\bar{\boldsymbol{x}}; f_{\mathrm{b}})$ and reinstate the constant factor $\frac{\bar{\psi}^{|\bar{\boldsymbol{x}}|}}{\psi^{|\boldsymbol{x}|}}$ and term in parentheses later.

Let $\mathcal{F} = \{\bar{f} \mid p_y(\boldsymbol{x}; \bar{f}) = \mu\}$ be the set of base classifiers that are consistent with the observed value of the smoothed class probability $\mu = p_y(\boldsymbol{x}; f_{\mathrm{b}})$. We seek an upper bound on $\sigma(\bar{\boldsymbol{x}}, f_{\mathrm{b}})$ that does not depend on the functional form of the base classifier, so we consider the worst case in the set $\mathcal{F}$:

$$\sigma(\bar{\boldsymbol{x}}; f_{\mathrm{b}}) \leq \max_{\bar{f} \in \mathcal{F}} \sum_{\boldsymbol{\epsilon} \in \mathcal{E}(\boldsymbol{x}, \boldsymbol{\epsilon}^\star)} \rho(\psi, \bar{\psi})^{|\boldsymbol{\epsilon}|} s_{\bar{f}}(\boldsymbol{\epsilon}, \boldsymbol{x}). \tag{14}$$

Rewriting the sum over $\mathcal{E}(\boldsymbol{x}, \boldsymbol{\epsilon}^\star)$ as a double sum over the number of retained elements $i$ and edits in $\mathcal{E}_i(\boldsymbol{x}, \boldsymbol{\epsilon}^\star)$ (see proof of Lemma 3), we find:

$$\text{RHS of (14)} = \max_{\bar{f} \in \mathcal{F}} \sum_{i=0}^{|\boldsymbol{z}^\star|} \rho(\psi, \bar{\psi})^i \sum_{\boldsymbol{\epsilon} \in \mathcal{E}_i(\boldsymbol{x}, \boldsymbol{\epsilon}^\star)} s_{\bar{f}}(\boldsymbol{\epsilon}, \boldsymbol{x})$$

$$= \max_{\bar{f} \in \mathcal{F}} \sum_{i=0}^{|\boldsymbol{z}^\star|} \psi^{|\boldsymbol{z}^\star|} \bar{\psi}^{-i} (1 - \bar{\psi})^i \sum_{\boldsymbol{\epsilon} \in \mathcal{E}_i(\boldsymbol{x}, \boldsymbol{\epsilon}^\star)} \mathbf{1}_{\bar{f}(\mathrm{apply}(\boldsymbol{x}, \boldsymbol{\epsilon}))=y}. \tag{15}$$

Let $[n] = \{0, \ldots, n\}$ denote the set of non-negative integers up to $n$. We can rewrite the maximization problem in Equation 15 as a maximization problem over the set of functions $\mathcal{H}$ from $[N]$ to $[M]$ where $N = |\boldsymbol{z}^\star|$ and $M = \bigcup_{i \in [N]} \binom{N}{i}$. Specifically, we have

$$\max_{h \in \mathcal{H}} \quad \sum_{i=0}^{N} v(i) h(i)$$

$$\text{s.t.} \quad h(i) \equiv \sum_{\boldsymbol{\epsilon} \in \mathcal{E}_i(\boldsymbol{x}, \boldsymbol{\epsilon}^\star)} \mathbf{1}_{\bar{f}(\text{apply}(\boldsymbol{x}, \boldsymbol{\epsilon})) = y} \quad \text{and} \quad \bar{f} \in \mathcal{F}, \tag{16}$$

where we have defined $v(i) := \psi^N \bar{\psi}^{-i} (1 - \bar{\psi})^i$. Next, we continue to lower bound the solution to the above problem by relaxing the constraints. First, we replace $\mathcal{F}$ by a superset:

$$\mathcal{F}' = \left\{ \bar{f} \ \Big| \ \sum_{i=0}^{N} w(i) \sum_{\boldsymbol{\epsilon} \in \mathcal{E}_i(\boldsymbol{x}, \boldsymbol{\epsilon}^\star)} \mathbf{1}_{\bar{f}(\text{apply}(\boldsymbol{x}, \boldsymbol{\epsilon})) = y} \leq \mu \right\},$$

where we have defined $w(i) := (1 - \psi)^i \psi^{N-i}$. The new constraint on $h$ (replacing Equation 16) is then

$$h(i) \equiv \sum_{\boldsymbol{\epsilon} \in \mathcal{E}_i(\boldsymbol{x}, \boldsymbol{\epsilon}^\star)} \mathbf{1}_{\bar{f}(\text{apply}(\boldsymbol{x}, \boldsymbol{\epsilon})) = y} \quad \text{and} \quad \sum_{i=0}^{N} w(i) h(i) \leq \mu \quad \text{and} \quad \bar{f} \in \mathcal{X} \to \mathcal{Y}.$$

Second, we relax the constraint further, by allowing $h(i)$ to be any integer in the set $[\binom{N}{i}]$.

The resulting relaxed maximization problem is a bounded knapsack problem:

$$\max_{h \in \mathcal{H}} \quad \sum_{i=0}^{N} v(i) h(i) \tag{17}$$

$$\text{s.t.} \quad \sum_{i=0}^{N} w(i) h(i) \leq \mu \quad \text{and} \quad h(i) \leq \binom{N}{i}. \tag{18}$$

Noting that the value per unit weight $v(i)/w(i) = \rho(\psi, \bar{\psi})^i$ is decreasing in $i$ for $\psi \leq \bar{\psi}$, we propose the following greedy "solution", which slightly violates the constraint:

$$h^*(i) = \begin{cases} \binom{N}{i}, & i < H^* \\ \left\lceil \frac{\mu - \sum_{j=0}^{H^*-1} \binom{N}{i} w(j)}{w(H^*)} \right\rceil, & i = H^* \\ 0, & \text{otherwise}, \end{cases}$$

where $H^* = \min_{H : \sum_{i=0}^{H} \binom{N}{i} w(j) \geq \mu} H$.

We now show that $h^*$ yields a valid upper bound on Equation 17. First, observe that

$$\sum_{i=0}^{N} w(i) h^*(i) \geq \mu, \tag{19}$$

which means the constraint in Equation 18 is not necessarily satisfied by $h^*$. Now suppose there exists a solution $h'$ to Equation 17 that yields a tighter upper bound than $h^*$. Define $\Delta(i) := w(i)(h^*(i) - h'(i))$ and observe that $\Delta(i) \geq 0$ when $i < H^*$ and $\Delta(i) \leq 0$ when $i > H^*$. By combining Equation 19 (for $h^*$) and Equation 18 (for $h'$), we have $\sum_{i=0}^{N} \Delta(i) \geq 0$. Together, these results imply

$$\sum_{i=0}^{N} \rho(\psi, \bar{\psi})^i \Delta(i) \geq \rho(\psi, \bar{\psi})^{H^*} \sum_{i=0}^{N} \Delta(i) \geq 0.$$

The first inequality holds because we are decreasing the magnitude of positive terms ($i < H^*$) and increasing the magnitude of negative terms ($i > H^*$), noting that $\rho(\psi, \bar{\psi})^i$ is decreasing in $i$. This then implies (by expanding the definition of $\Delta(i)$):

$$\sum_{i=0}^{N} v(i) h^*(i) = \sum_{i=0}^{N} \rho(\psi, \bar{\psi})^i w(i) h^*(i) \geq \sum_{i=0}^{N} \rho(\psi, \bar{\psi})^i w(i) h'(i) = \sum_{i=0}^{N} v(i) h'(i),$$

which is a contradiction, thereby confirming that $h^*$ yields a valid upper bound.

Substituting $h^*$ in the objective yields the following upper bound:

$$\sigma(\bar{\boldsymbol{x}}; f_{\mathrm{b}}) \leq \sum_{i=0}^{N} v(i) h^*(i)$$

$$= \left(\frac{\psi}{\bar{\psi}}\right)^{|\boldsymbol{z}^\star|} \left\{ \sum_{i=0}^{H^*-1} \mathcal{B}_i(|\boldsymbol{z}^\star|, \bar{\psi}) + \mathcal{B}_{H^*}(|\boldsymbol{z}^\star|, \bar{\psi}) \left\lfloor \frac{\mu - \sum_{j=0}^{H^*-1} \mathcal{B}_j(|\boldsymbol{z}^\star|, \psi)}{\mathcal{B}_{H^*}(|\boldsymbol{z}^\star|, \psi)} \right\rfloor_{\binom{|\boldsymbol{z}^\star|}{H^*}^{-1}} \right\}.$$

Finally, we multiple both sides of the above inequality by $\frac{\bar{\psi}^{|\bar{\boldsymbol{x}}|}}{\psi^{|\boldsymbol{x}|}}$ to obtain the desired upper bound on $p_y(\bar{\boldsymbol{x}}; f_{\mathrm{b}})$. $\qquad\square$

**Theorem 4.** *Given a smoothed classifier $f$ as described in Algorithm 2 with a length-dependent deletion function $\psi$,*

$$\forall \bar{\boldsymbol{x}} \in B_r(\boldsymbol{x}; \mathsf{o}) : f(\boldsymbol{x}) = f(\bar{\boldsymbol{x}})$$

*with confidence level at least $1 - \alpha$.*

*Proof.* By the definition of the smoothed classifier $f$, the prediction is determined by the probability distribution induced by $\phi_{\psi(\boldsymbol{x})}$. Let $t_1^{\mathrm{lb}}$ and $t_2^{\mathrm{ub}}$ be the Clopper-Pearson lower and upper confidence bounds, respectively, for the probabilities of the top class $y_1$ and the runner-up class $y_2$. By Algorithm 1, these are estimated such that, with confidence at least $1 - \alpha$,

$$p_{y_1}(\boldsymbol{x}; f_{\mathrm{b}}, \phi_{\psi(\boldsymbol{x})}) \geq t_1^{\mathrm{lb}} \quad \text{and} \quad t_2^{\mathrm{ub}} \geq p_{y_2}(\boldsymbol{x}; f_{\mathrm{b}}, \phi_{\psi(\boldsymbol{x})}).$$

For any perturbation $\bar{\boldsymbol{x}} \in B_r(\boldsymbol{x}; \mathsf{o})$, Algorithm 1 and Lemma 3 ensures that

$$t_1^{\mathrm{lb}} \geq \bar{t}_1^{\mathrm{lb}} \geq \bar{t}_2^{\mathrm{ub}} \geq t_2^{\mathrm{ub}}.$$

By transitivity, it follows that

$$p_{y_1}(\boldsymbol{x}; f_{\mathrm{b}}, \phi_{\psi(\boldsymbol{x})}) > p_{y_2}(\boldsymbol{x}; f_{\mathrm{b}}, \phi_{\psi(\boldsymbol{x})})$$

with probability at least $1 - \alpha$. Hence, the predicted class remains $y_1$ for all perturbations within $B_r(\boldsymbol{x}; \mathsf{o})$, completing the proof. $\qquad\square$

**Lemma 7.** *By setting the deletion probability as a fixed constant, i.e., $\psi(\boldsymbol{x}) := p_{\mathsf{del}}$ for all $\boldsymbol{x}$, Lemma 3 recovers Theorem 5 of Huang et al. [17] with only a negligible difference.*

*Specifically, let $\boldsymbol{x}, \bar{\boldsymbol{x}} \in \mathcal{X}$ be a pair of inputs with longest common subsequence (LCS) $\boldsymbol{z}^\star$ and $\mu = p_y(\boldsymbol{x}, f_{\mathrm{b}})$. Under this assumption, the probability $p_y(\bar{\boldsymbol{x}}; f_{\mathrm{b}})$ satisfies the lower bound*

$$p_y(\bar{\boldsymbol{x}}; f_{\mathrm{b}}) \geq p_{\mathsf{del}}^{|\bar{\boldsymbol{x}}|-|\boldsymbol{x}|} \left( \mu - 1 + p_{\mathsf{del}}^{|\boldsymbol{x}|-|\boldsymbol{z}^\star|} - (1 - p_{\mathsf{del}})^{H^*} p_{\mathsf{del}}^{|\boldsymbol{z}^\star|-H^*} \right).$$

*Here, $H^*$ represents the minimal number of retained tokens required to satisfy*

$$\sum_{i=0}^{H^*} \mathcal{B}_i(|\boldsymbol{z}^\star|, p_{\mathsf{del}}) \geq \mu - 1 + p_{\mathsf{del}}^{|\boldsymbol{x}|-|\boldsymbol{z}^\star|},$$

*where $\mathcal{B}_k(n, p) := \binom{n}{k}(1 - p)^k p^{n-k}$ is the Binomial pmf for $n$ trials with with success probability $1 - p$.*

*The difference from Theorem 5 of Huang et al. [17] is solely due to the term $(1 - p_{\mathsf{del}})^{H^*} p_{\mathsf{del}}^{|\bar{\boldsymbol{x}}|-|\boldsymbol{x}|+|\boldsymbol{z}^\star|-H^*}$, which is typically less than $10^{-7}$, negligible compared to the other terms.*

*Proof.* As $\psi = \bar{\psi} = p_{\mathsf{del}}$,

$$
p_y(\bar{\boldsymbol{x}}; f_{\mathsf{b}}) \geq \frac{\bar{\psi}^{|\bar{\boldsymbol{x}}| - |\boldsymbol{z}^\star|}}{\psi^{|\boldsymbol{x}| - |\boldsymbol{z}^\star|}} \left\{ \sum_{i=\boldsymbol{l}(H^*-1)}^{(1-\boldsymbol{l})(H^*-1)+\boldsymbol{l}|\boldsymbol{z}^\star|} \mathcal{B}_i(|\boldsymbol{z}^\star|, \bar{\psi}) \right.
$$

$$
\left. + \mathcal{B}_{H^*}(|\boldsymbol{z}^\star|, \bar{\psi}) \left\lfloor \frac{\mu - 1 + \psi^{|\boldsymbol{x}| - |\boldsymbol{z}^\star|} - \sum_{i=\boldsymbol{l}(H^*-1)}^{(1-\boldsymbol{l})(H^*-1)+\boldsymbol{l}|\boldsymbol{z}^\star|} \mathcal{B}_j(|\boldsymbol{z}^\star|, \psi)}{\mathcal{B}_{H^*}(|\boldsymbol{z}^\star|, \psi)} \right\rfloor_{\binom{|\boldsymbol{z}^\star|}{H^*}^{-1}} \right\}
$$

$$
= p_{\mathsf{del}}^{|\bar{\boldsymbol{x}}| - |\boldsymbol{x}|} \left\{ \sum_{i=0}^{H^*-1} \mathcal{B}_i(|\boldsymbol{z}^\star|, p_{\mathsf{del}}) \right.
$$

$$
\left. + \mathcal{B}_{H^*}(|\boldsymbol{z}^\star|, p_{\mathsf{del}}) \left\lfloor \frac{\mu - 1 + p_{\mathsf{del}}^{|\boldsymbol{x}| - |\boldsymbol{z}^\star|} - \sum_{i=0}^{H^*-1} \mathcal{B}_j(|\boldsymbol{z}^\star|, p_{\mathsf{del}})}{\mathcal{B}_{H^*}(|\boldsymbol{z}^\star|, p_{\mathsf{del}})} \right\rfloor_{\binom{|\boldsymbol{z}^\star|}{H^*}^{-1}} \right\}
$$

$$
\geq p_{\mathsf{del}}^{|\bar{\boldsymbol{x}}| - |\boldsymbol{x}|} \left\{ \sum_{i=0}^{H^*-1} \mathcal{B}_i(|\boldsymbol{z}^\star|, p_{\mathsf{del}}) \right.
$$

$$
\left. + \mathcal{B}_{H^*}(|\boldsymbol{z}^\star|, p_{\mathsf{del}}) \left( \frac{\mu - 1 + p_{\mathsf{del}}^{|\boldsymbol{x}| - |\boldsymbol{z}^\star|} - \sum_{i=0}^{H^*-1} \mathcal{B}_j(|\boldsymbol{z}^\star|, p_{\mathsf{del}})}{\mathcal{B}_{H^*}(|\boldsymbol{z}^\star|, p_{\mathsf{del}})} - \binom{|\boldsymbol{z}^\star|}{H^*}^{-1} \right) \right\}
$$

$$
= p_{\mathsf{del}}^{|\bar{\boldsymbol{x}}| - |\boldsymbol{x}|} \left\{ \mu - 1 + p_{\mathsf{del}}^{|\boldsymbol{x}| - |\boldsymbol{z}^\star|} \right\} - (1 - p_{\mathsf{del}})^{H^*} p_{\mathsf{del}}^{|\bar{\boldsymbol{x}}| - |\boldsymbol{x}| + |\boldsymbol{z}^\star| - H^*}
$$

The first equality holds because $\boldsymbol{l}$ evaluates to 0. The last inequality holds because $\lfloor a \rfloor \geq a - 1$, which applies to the gridded flooring operation.

This result closely matches Theorem 5 of Huang et al. [17], differing only by the term $(1 - p_{\mathsf{del}})^{H^*} p_{\mathsf{del}}^{|\bar{\boldsymbol{x}}| - |\boldsymbol{x}| + |\boldsymbol{z}^\star| - H^*}$, which is typically less than $10^{-7}$, negligible compared to the other terms. $\qquad\square$

## B    Certification for Arbitrary Deletion Probability Functions

In Lemmas 3 and 6, we obtained bounds on the smoothed class probabilities at a neighboring input $\bar{\boldsymbol{x}}$ to some query input $\boldsymbol{x}$. These bounds can be used directly for edit distance certification, by exhaustively enumerating over all neighboring inputs $\bar{\boldsymbol{x}}$ that are within edit distance $r$ from $\boldsymbol{x}$. Algorithm 2 details how this can be done. We note that this algorithm provides a significant advantage over naively certifying the smoothed model. A naive approach would require running a full Monte Carlo estimation for every neighbor $\bar{\boldsymbol{x}}$, whereas our algorithm performs the costly estimation only once for the original input $\boldsymbol{x}$. The certification check for each neighbor then relies on the analytical bounds, which are computationally inexpensive.

---

**Algorithm 2** CERTIFYGENERAL

---

**Require:** $f_{\mathsf{b}}$: base classifier, $\boldsymbol{x}$: input sequence, $y_1$: predicted class, $\psi$: length-dependent deletion probability function, o: allowed edit operations, $\alpha$: significance level
**Ensure:** $r$: maximum radius that can be certified
1:  $t_1^{\mathsf{lb}} \leftarrow \hat{p}_{y_1}^{\mathsf{lb}}(\boldsymbol{x}; f_{\mathsf{b}}, \phi_\psi, \alpha)$
2:  $t_2^{\mathsf{ub}} \leftarrow \max_{y \neq y_1} \hat{p}_y^{\mathsf{ub}}(\boldsymbol{x}; f_{\mathsf{b}}, \phi_\psi, \alpha)$
3:  **for** $r = 0$ to $\infty$ **do**
4:      **for all** $\bar{\boldsymbol{x}} \in B_{r+1}(\boldsymbol{x}; \mathrm{o})$ **do**
5:          $\bar{t}_1^{\mathsf{lb}} \leftarrow \mathrm{lb}(t_1^{\mathsf{lb}}, \boldsymbol{x}, \bar{\boldsymbol{x}}, \psi)$
6:          $\bar{t}_2^{\mathsf{ub}} \leftarrow \mathrm{ub}(t_2^{\mathsf{ub}}, \boldsymbol{x}, \bar{\boldsymbol{x}}, \psi)$
7:          **if** $\bar{t}_1^{\mathsf{lb}} \leq \bar{t}_2^{\mathsf{ub}}$ **then**
8:              **return** r
9:          **end if**
10:     **end for**
11: **end for**

---

# C  AdaptDel+ Details and Intuition

In this section, we provide details and intuition behind the optimization process for AdaptDel+. The core idea is to calibrate the deletion mechanism by finding optimal expected lengths for each input bin, ensuring that certified robustness is maximized while maintaining a minimum certified accuracy threshold.

**Create Equal Width Bins**  Algorithm 3 is used to create bins based on the lengths of the input sequences, ensuring that each bin contains at least a minimum number of samples. The algorithm first computes the lengths of all sequences in the dataset and trims extreme outliers based on a specified percentage. It then iteratively reduces the target number of bins, recalculating a new set of equally spaced boundaries in each step, until all bins contain at least the minimum number of samples. The result is a set of bin boundaries that divides the dataset into equal-width intervals, tailored to the data distribution while avoiding sparsity issues. We use these crude bin boundaries with some manual calibration. We report the exact bin boundaries used in Appendix D.

---

**Algorithm 3** CREATEDYNAMICEQUALWIDTHBINS

---

**Require:** Dataset $\mathbb{D}$, Threshold $C$, Outlier percentage $\alpha$ (default: $1\%$)
**Ensure:** Bin boundaries $B$
 1: Compute input lengths for all samples in $\mathbb{D}$
 2: Trim lengths outside $\alpha$ and $100 - \alpha$ percentiles
 3: Initialize $k \leftarrow$ estimated number of bins, $B_k \leftarrow$ equally spaced boundaries
 4: **while** True **do**
 5:    Assign data to bins defined by $B_k$
 6:    **if** all bins have at least $C$ samples **then**
 7:       **Break**
 8:    **end if**
 9:    $k \leftarrow k - 1$
10:    Update $B \leftarrow$ equally spaced boundaries for $k$ bins
11:    **if** $k < 1$ **then**
12:       **Raise** error: Cannot satisfy threshold
13:    **end if**
14: **end while**
15: **Return** $B_k$

---

**Optimizing Certified Radius**  Algorithm 4 describes the process of determining the maximum certified radius $(r)$ that meets a specific certified accuracy threshold $(\tau)$. This is achieved by iteratively calculating the certified accuracy for each radius using the dataset and checking whether it satisfies the threshold. The objective is to return the largest $r$ and the corresponding certified accuracy $(\text{CertAcc}_r)$ for that level of robustness. However, due to the discrete nature of the objective, the optimization surface is non-smooth, making direct optimization challenging. To address this, we also calculate the certified accuracy, which provides a smoother metric helpful for hyperparameter tuning.

---

**Algorithm 4** MAXCERTRADIUS

---

**Require:** $f_{\text{b}}$: Base model, $k$: Expected lengths after deletion, $\mathbb{D}$: Dataset, $\tau$: Certified accuracy threshold, $\mathsf{o}$: allowed edit operations
**Ensure:** Maximum certified radius $r$, Certified accuracy $\text{CertAcc}_r$
 1: Define deletion function $\psi(\boldsymbol{x}) := 1 - \frac{k}{|\boldsymbol{x}|}$
 2: $f_{\text{CR}}(\boldsymbol{x}) := \text{CERTIFY}(f_{\text{b}}, \boldsymbol{x}, \psi, \mathsf{o})$
 3: **for** $r = 0$ to $\infty$ **do**
 4:    $\text{CertAcc}_r \leftarrow \sum_{(\boldsymbol{x},y) \in \mathbb{D}} \frac{\mathbf{1}_{f(\boldsymbol{x})=y} \mathbf{1}_{f_{\text{CR}}(\boldsymbol{x}) \geq r}}{|\mathbb{D}|}$
 5:    **if** $\text{CertAcc}_r < \tau$ **then**
 6:       **return** $r - 1, \text{CertAcc}_{r-1}$
 7:    **end if**
 8: **end for**

---

**Finding Optimal Expected Lengths** The AdaptDel+ mechanism employs Algorithm 5 to optimize the expected lengths for each input bin. The input data is divided into bins based on sequence lengths, and for each bin, we identify the expected length after deletion that maximizes the certified radius and accuracy while adhering to the minimum accuracy threshold. The relationship between the expected length and the optimization objectives is approximately unimodal, allowing the use of efficient search methods.

We adopt golden section search to minimize the number of queries required for optimization. This choice is particularly advantageous as computing certified radii and accuracies involves expensive stochastic sampling from randomized smoothed models. Golden section search strikes a balance between accuracy and computational efficiency, making it well-suited for this problem. The asymptotic complexity of this calibration process is linear in both the number of bins, $n$, and the sample size per bin, $m$. This results from the main loop iterating through each of the $n$ bins, while the optimization search performed within each bin has a cost that is linear in the $m$ samples being evaluated.

---

**Algorithm 5** OPTIMIZEEXPECTEDLENGTH

---

**Require:** $f_{\mathrm{b}}$: base model, $\mathbb{D}$: training dataset, $\{g_0, g_1, \ldots, g_n\}$: bin intervals, $\tau$: certified accuracy threshold, tol: search tolerance, $m$: sample size per bin, o: allowed edit operations
**Ensure:** Optimized expected lengths $K$
1: Initialize $K \leftarrow [0, \ldots, 0]$
2: **for** $i \in \{1, \ldots, n-1\}$ **do**
3:     Sample $m$ data points from the interval
            $\mathbb{D}_i \leftarrow \{\boldsymbol{x} \mid |\boldsymbol{x}| \in [g_i, g_{i+1})\}$
4:     low $\leftarrow 0.01 \cdot g_{i+1}$, high $\leftarrow 0.3 \cdot g_i$
5:     **while** high $-$ low $>$ tol **do**
6:         $m_1 \leftarrow$ low $+ (3 - \sqrt{5})/2 \cdot ($high $-$ low$)$
7:         $m_2 \leftarrow$ high $- (3 - \sqrt{5})/2 \cdot ($high $-$ low$)$
8:         $r_1, \mathrm{CA}_1 \leftarrow$ MAXCERTRADIUS$(f_{\mathrm{b}}, m_1, \mathbb{D}, \tau, \mathsf{o})$
9:         $r_2, \mathrm{CA}_2 \leftarrow$ MAXCERTRADIUS$(f_{\mathrm{b}}, m_2, \mathbb{D}, \tau, \mathsf{o})$
10:        **if** $(r_1 > r_2)$ **or** $(r_1 = r_2$ **and** $\mathrm{CA}_1 > \mathrm{CA}_2)$ **then**
11:            high $\leftarrow m_2$
12:        **else**
13:            low $\leftarrow m_1$
14:        **end if**
15:    **end while**
16:    $K[i] \leftarrow ($high $+$ low$)/2$
17: **end for**
18: **return** Optimized expected lengths $K$

---

**Stochastic Challenges** Despite its efficiency, the optimization process can be sensitive to the stochastic nature of randomized smoothing models and potential biases in binned data. This occasionally results in optimized expected lengths ($k$) that are not strictly increasing with bin lengths, which can slightly affect the smoothness of the calibration. To mitigate this, additional regularization or smoothing steps may be considered during the optimization process.

The resulting optimized expected lengths ($K$) are then used to parameterize the deletion function $\psi(\boldsymbol{x})$, as defined in Section 3:

$$\psi(\boldsymbol{x}) = 1 - \frac{K_{g(\boldsymbol{x})}}{|\boldsymbol{x}|}$$

where $g(\boldsymbol{x})$ is the index of the bin corresponding to the input $\boldsymbol{x}$. This adaptive approach ensures robust performance across varying input lengths and scenarios.

# D   Evaluation Setup

We describe the detailed evaluation setup in this section.

| Dataset | Avg. Words | Number of Instances | | |
|---|---|---|---|---|
| | | Train | Valid | Test[*] |
| Yelp | 134.1 | 585 000 | 65 000 | 10 000 |
| SpamAssassin | 228.2 | 2 152 | 239 | 2 378 |
| IMDB | 231.2 | 22 500 | 2 500 | 25 000 |
| LUN | 269.9 | 13 416 | 1 490 | 6 454 |

Table 1: Summary of datasets. "Avg. Words" denotes the average number of words per instance in the dataset.
[*]The full test set is used for evaluation for all datasets except for Yelp, where we sample $10\,000$ instances from the available $50\,000$ instances.

## D.1 Dataset Specification

We collect all data from HuggingFace Datasets[1] and the `AdvBench` repository[2] [31].

## D.2 Parameter Settings

| | Parameter | Values |
|---|---|---|
| **Base model** | Model | `AutoModelForSequenceClassification("roberta-base")` |
| | Tokenizer | `AutoTokenizer("roberta-base")` |
| **Scheduler** | Python command | `transformers.get_linear_schedule_with_warmup` |
| | Warmup epochs | 10 |
| **Optimizer** | Python class | `torch.optim.AdamW` |
| | Learning rate | 2.0E-5 |
| | Weight decay | 1.0E-6 |
| | Gradient clipping | `clip_grad_norm_(model.parameters(), 1.0)` |
| **Training** | Batch size | 32 |
| | Max. epoch | 200 |
| | Early stopping | No improvement in validation loss after 25 epochs |

Table 2: Parameter settings for RoBERTa, the optimizer and training procedure. Parameter settings are consistent across all models (NoSmooth, CERT-ED, RanMASK, AdaptDel, AdaptDel+) except where specified.

For all randomized smoothing mechanisms, perturbations are applied at the word level. Specifically, an input text is first split into a sequence of words using white-space tokenization. The smoothing mechanism then deletes words from this sequence. The resulting sequence of words is joined back into a string, which is then passed to the base RoBERTa model's tokenizer for processing into subword tokens. We fine-tune the base RoBERTa model on these perturbed inputs using the respective training split for each dataset. The default parameter configurations for our experiments are provided in Table 2. We avoid individually calibrating the optimizer or training schedule for each model, as the default settings demonstrate robust performance across all datasets. To approximate the smoothed models (RanMASK, CERT-ED, AdaptDel, AdaptDel+), we rely on Monte Carlo sampling with 1000 perturbed inputs for prediction and 4000 for certification, using a significance level of 0.05.

## D.3 AdaptDel+ Setup

To train and calibrate AdaptDel+, we use the parameter settings provided in Table 3. Since the optimized expected lengths are not available during training, we use a random deletion probability ranging from $70\%$ to $99\%$. Empirically, this strategy proved effective in producing a stable smoothed classifier. The certified accuracy threshold ($\tau$) is set to $40\%$ for the Yelp dataset and $75\%$ for the

---

[1]`https://github.com/huggingface/datasets`
[2]`https://github.com/thunlp/Advbench`

| Dataset | Parameter | Values |
|---|---|---|
| Yelp | Bin boundaries $B$ | $[0, 28, 82, 135, 189, 243, 296, 350, 404,$ $457, 511, 565, 619, 672, 726, 780, 887, \infty]$ |
| | Threshold $\tau$ | 0.40 |
| | Sample size per bin | 200 |
| IMDB | Bin boundaries $B$ | $[0, 72, 133, 194, 256, 318, 379, 440, 502,$ $564, 625, 686, 748, 810, 871, 932, \infty]$ |
| | Threshold $\tau$ | 0.75 |
| | Sample size per bin | 200 |
| SpamAssassin | Bin boundaries $B$ | $[0, 137, 230, 324, \infty]$ |
| | Threshold $\tau$ | 0.75 |
| | Sample size per bin | 50 |
| LUN | Bin boundaries $B$ | $[0, 85, 150, 216, 282, 347, \infty]$ |
| | Threshold $\tau$ | 0.75 |
| | Sample size per bin | 200 |

Table 3: Parameters used in optimizing AdaptDel+.

IMDB, SpamAssassin, and LUN datasets. Bins are created using the entire training set (excluding validation samples), and manual deletion of boundaries are made if the number of bins is excessive. For optimizing the expected lengths, we use Monte Carlo sampling with a prediction size of 32 and a certification size of 256. The specific number of samples per bin used for optimization is outlined in Table 3. Finally, we do not retrain the base model after determining the optimal deletion rates.

# E   Additional Experimental Results

We provide additional experimental results and details in this section. We first present the values of $\psi(\boldsymbol{x})$ and $k_{g(\boldsymbol{x})}$ for AdaptDel and AdaptDel+ in Appendix E.1. We then provide additional certified accuracy results, the quantile certified accuracy results for the IMDB and LUN datasets, and certified accuracy as a function of the certified radius for all datasets in Appendix E.2. We also include an ablation study on the $p_{\text{del}}$ values for AdaptDel and AdaptDel+ in Appendix E.3. Finally, we conclude with a discussion on the computation complexity and cost of our methods in Appendix F.1.

## E.1   Parameters for AdaptDel and AdaptDel+

We present the values of $\psi(\boldsymbol{x})$ and $k_{g(\boldsymbol{x})}$ for AdaptDel and AdaptDel+ in Figure 3. Unlike Adapt-Del, AdaptDel+ does not enforce a strictly monotonically increasing $\psi(\boldsymbol{x})$. Instead, AdaptDel+ dynamically adjusts $\psi(\boldsymbol{x})$ according to the characteristics of the dataset. Future work could further investigate richer classes of $\psi(\boldsymbol{x})$, potentially including regularization methods. Our framework is specifically designed to allow such flexibility.

## E.2   Other Certified Accuracy Results

As illustrated in Table 4, both AdaptDel and AdaptDel+ maintain competitive clean accuracy while significantly enhancing robustness guarantees compared to RanMASK and CERT-ED. In particular, they improved the mean certified radius by an average of over $50\%$, and the median cardinality of the certified region by up to *30 orders of magnitude*. This substantial expansion highlights the effectiveness of adaptive deletion in certifying robustness against a significantly broader space of perturbations, making it particularly valuable in adversarial settings where sequence manipulations are common.

We report standard deviation (normalised as standard error) in Table 4, but note that it does not directly imply statistical significance due to input length being a key variable. Instead, we report

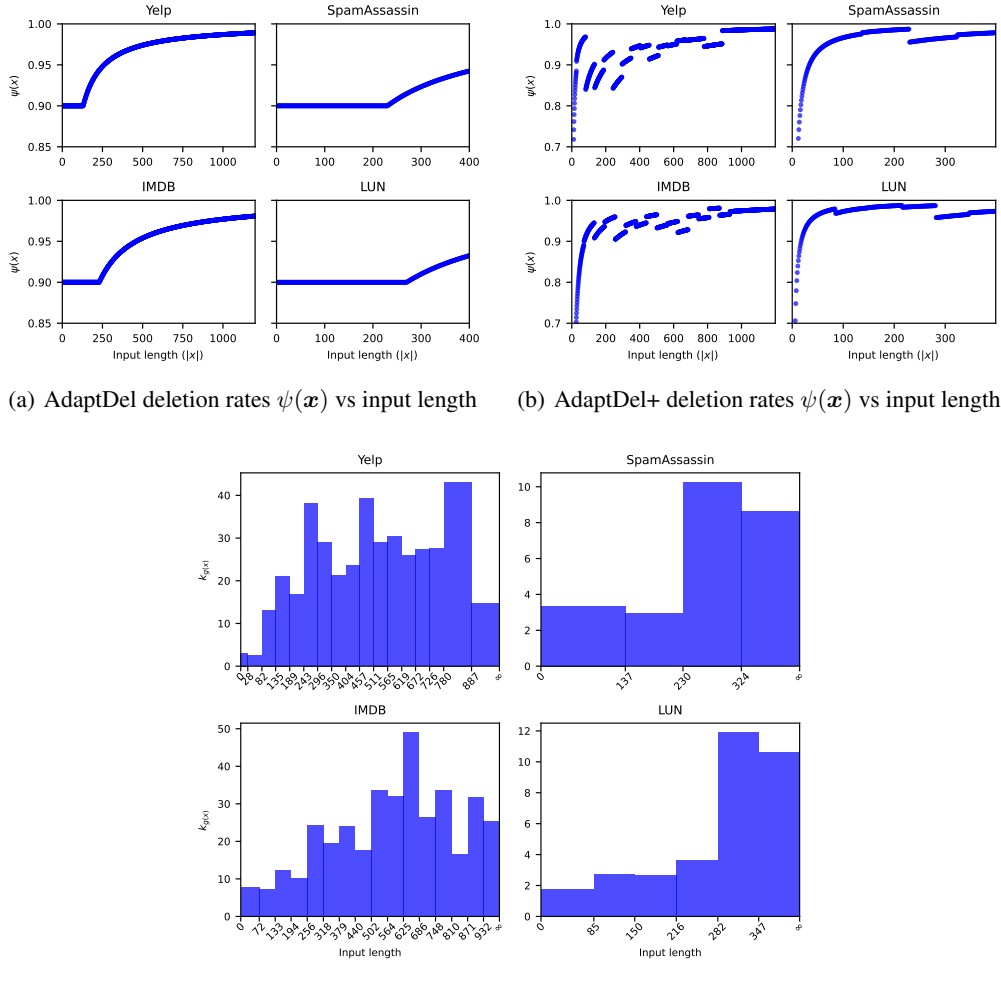

(a) AdaptDel deletion rates $\psi(\boldsymbol{x})$ vs input length

(b) AdaptDel+ deletion rates $\psi(\boldsymbol{x})$ vs input length

(c) AdaptDel+ retention rates $k_{g(\boldsymbol{x})}$ vs input length

Figure 3: Plots of the deletion rate $\psi(\boldsymbol{x})$ for AdaptDel and AdaptDel+ and the retention length $k_{g(\boldsymbol{x})}$ for AdaptDel+. The top left plot shows the deletion rates $\psi(\boldsymbol{x})$ for AdaptDel as a function of input length. The top right plot shows the deletion rates $\psi(\boldsymbol{x})$ for AdaptDel+ as a function of input length. The bottom left plot shows the retention rates $k_{g(\boldsymbol{x})}$ for AdaptDel+ as a function of input length. The plots demonstrate that AdaptDel and AdaptDel+ adaptively adjust their deletion rates based on the input length.

| Dataset | Model | Clean Acc. | Mean CR | Wass. Dist. | Agg. log(CC) |
|---|---|---|---|---|---|
| Yelp | NoSmooth | 69.25 | — | — | — |
| | RanMASK, 90% | 57.15 | $0.61 \pm 0.013$ | 0.31 | 0.00 |
| | CERT-ED, 90% | 58.57 | $0.75 \pm 0.013$ | 0.26 | **7.16** |
| | AdaptDel, 90% | 56.98 | $\mathbf{0.99} \pm 0.023$ | 0.28 | 6.89 |
| | AdaptDel+ | 56.22 | $0.77 \pm 0.016$ | **0.24** | 6.78 |
| SpamAssassin | NoSmooth | 93.48 | — | — | — |
| | RanMASK, 90% | 86.87 | $2.33 \pm 0.012$ | 0.43 | 14.02 |
| | CERT-ED, 90% | 88.26 | $2.35 \pm 0.012$ | 0.42 | 14.49 |
| | AdaptDel, 90% | 87.99 | $\mathbf{2.86} \pm 0.018$ | **0.20** | **14.77** |
| | AdaptDel+ | 88.06 | $2.77 \pm 0.016$ | 0.24 | 14.01 |
| Yelp | NoSmooth | 98.02 | — | — | — |
| | RanMASK, 90% | 97.65 | $5.09 \pm 0.031$ | 0.58 | 37.49 |
| | CERT-ED, 90% | 97.81 | $5.03 \pm 0.031$ | 0.57 | 38.66 |
| | AdaptDel, 90% | 97.73 | $6.25 \pm 0.053$ | 0.30 | 38.79 |
| | AdaptDel+ | 96.93 | $\mathbf{12.35} \pm 0.152$ | **0.14** | **72.74** |
| LUN | NoSmooth | 99.16 | — | — | — |
| | RanMASK, 90% | 97.91 | $4.83 \pm 0.022$ | 0.31 | 34.51 |
| | CERT-ED, 90% | 98.20 | $4.94 \pm 0.019$ | 0.34 | 37.68 |
| | AdaptDel, 90% | 97.88 | $5.65 \pm 0.028$ | 0.30 | 38.90 |
| | AdaptDel+ | 95.09 | $\mathbf{10.69} \pm 0.088$ | **0.24** | **66.90** |

Table 4: Comparison of robustness certificates across models and datasets. All metrics are computed on the full test set. "Mean CR" refers to the mean certified radius, reported with standard error. A larger standard error indicates greater variability in certified radii across examples. We report the Wasserstein distance ("Wass. Dist.") between the standardized distributions of certified radius and input length as a measure of how well each method adapts to variations in input length (smaller is better). The right-most column reports the median log-cardinality of the certified region. For the Yelp dataset, the first quartile (Q1) of $\log(CC)$ is reported instead, since the median certified region cardinality is zero. Despite slight degradation in clean accuracy, both AdaptDel and AdaptDel+ significantly outperform RanMASK and CERT-ED in terms of both median and mean robustness.

the Wasserstein distance to capture how well each method adapts to variations in input length. Our adaptive methods achieve a lower Wasserstein distance, indicating a stronger alignment between the distribution of certified radii and the distribution of input lengths. This successful adaptation is also reflected in a higher variability (SE) of the certified radius, as the method appropriately assigns different radii to inputs of different lengths.

Figure 4 extends our analysis to the IMDB and LUN datasets, reinforcing the trends observed in Figure 2. In both datasets, certified accuracy improves with input size and log-cardinality, with adaptive deletion methods (AdaptDel, AdaptDel+) outperforming RanMASK and CERT-ED, especially for longer sequences. For IMDB, since more than $50\%$ of the length variation is concentrated in the last quartile, we observe a significant improvement in robustness in this region. In contrast, performance in the other three quartiles is more mixed, aligning with our previous observations on the Yelp dataset in Figure 2. For the LUN dataset, interestingly, AdaptDel+ underperforms in the first quartile, which may be due to the instability of the AdaptDel+ optimization procedure, as discussed in Appendix C. This suggests potential areas for improvement in AdaptDel+. However, its performance improves gradually with increasing input length, similar to the trend observed in SpamAssassin, where both AdaptDel and AdaptDel+ achieve significantly stronger robustness for longer sequences. These results further confirm the scalability of our approach across varying text distributions and input sizes.

Certified accuracy cannot be summarized by a single number without losing critical information about robustness under varying radii. To address this, we provide Figure 5, which plots certified accuracy as a function of the certification radius, as per convention. Note that this comparison disadvantages AdaptDel, AdaptDel+, and CERT-ED, as RanMASK operates under the more constrained Hamming distance threat model. Despite this, our methods consistently outperform others, following a similar pattern to the certified accuracy vs. log-cardinality plot. This further supports our use of log-

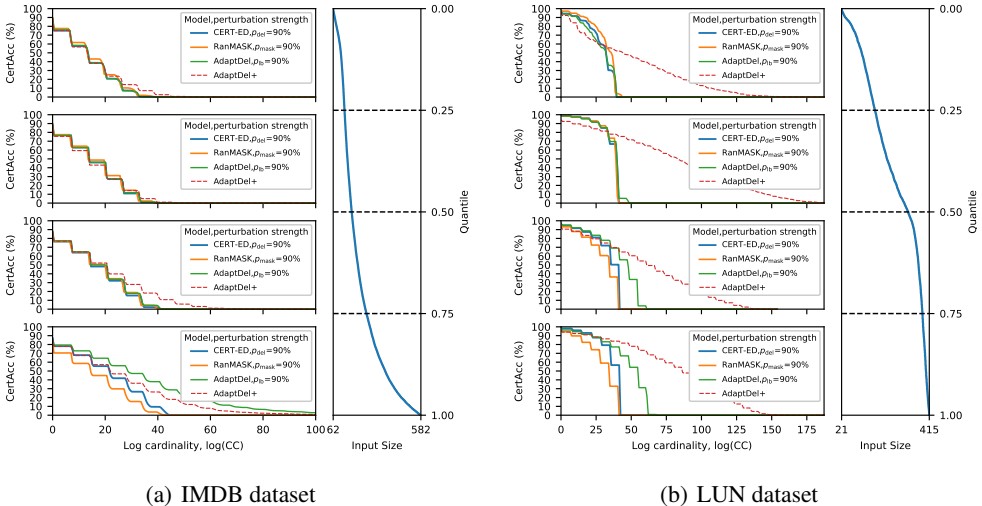

(a) IMDB dataset                         (b) LUN dataset

Figure 4: Certified accuracy as a function of the log-cardinality of the certified region, grouped by quartile of input size. The subfigure on the right displays the quantile by input size, with the dashed lines indicating the quartiles corresponding to the certified accuracy plots on the left. Each set of axes (top to bottom) corresponds to a split of the test set on the length-based quartiles (smallest to largest). For example, the second plot from top to bottom shows the certified accuracies of examples with in Q1 to Q2. The results demonstrate that the methods scale effectively across varying input sizes, with higher certified accuracy achieved for larger input sizes and higher log-cardinality regions.

cardinality certified accuracy as a balanced summary metric that does not unfairly favor any specific method.

### E.3   Ablation Study on Deletion Rate Lower Bound for AdaptDel

As shown in Table 5, increasing the lower bound $p_{lb}$ (and reducing the retention count $k = \lfloor (1 - p_{lb}) \mathbb{E}\{|\boldsymbol{x}|\} \rfloor$) in AdaptDel consistently improves the certified robustness across all datasets, albeit at a modest cost to clean accuracy. For instance, on the IMDB dataset, raising $p_{lb}$ from $50\%$ to $95\%$ enhances the mean certified radius from $0.34$ to $4.26$, with a corresponding increase in the aggregated log-cardinality from $0.00$ to $20.87$. Similar trends are observed on the SpamAssassin and LUN datasets, where higher $p_{lb}$ values lead to significant gains in both the mean certified radius and certified region size. This demonstrates that enforcing more aggressive deletion rates amplifies the certifiability of AdaptDel, especially on longer-input datasets. Furthermore, the Wasserstein distance between certified radius and input length steadily decreases with higher $p_{lb}$, suggesting that the certified radius becomes more input-length-aware as deletion strengthens. These results underscore the critical role of tuning deletion rates in balancing clean accuracy and robustness guarantees, enabling AdaptDel to scale effectively with input complexity.

## F   Analysis of Computational Efficiency

In this appendix, we document the computational requirements for training and certifying the models used in our study. We begin by benchmarking certification time for each method on the SpamAssassin dataset. Next, we report the compute costs of our remaining experiments (both training and certification) as presented in the main paper. Overall, we demonstrate that AdaptDel and AdaptDel+ match or exceed the efficiency of CERT-ED and RanMASK in both training and certification.

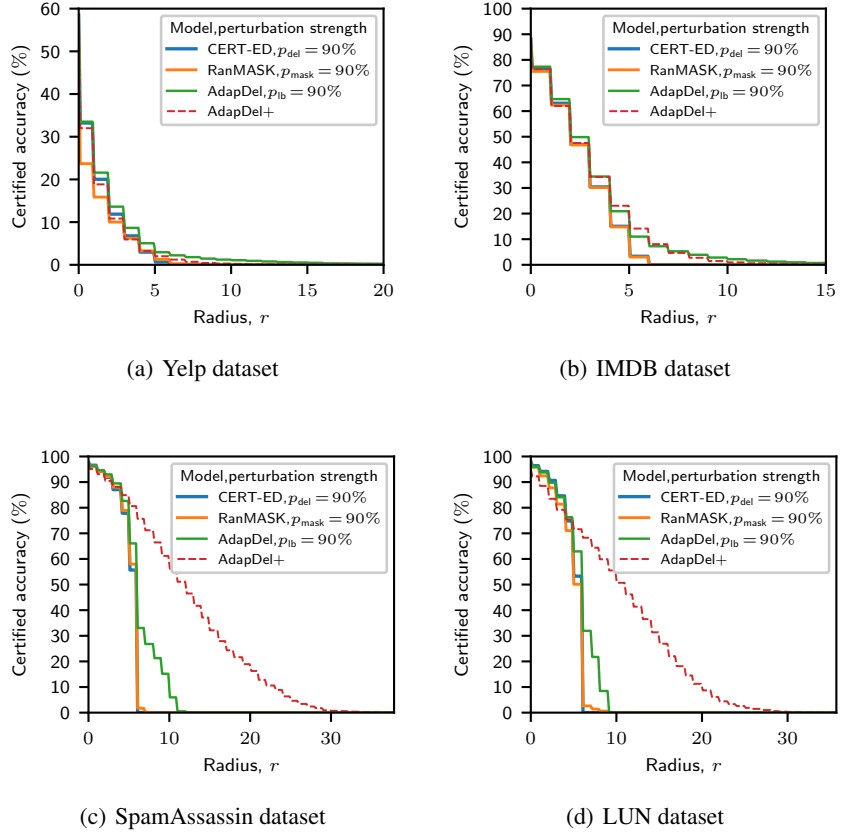

(a) Yelp dataset

(b) IMDB dataset

(c) SpamAssassin dataset

(d) LUN dataset

Figure 5: Certified accuracy for all methods as a function of the certified radius. While AdaptDel consistently outperforms RanMASK and CERT-ED across all radii, AdaptDel+ achieves the highest certified accuracy for larger certified regions. Note that, CERT-ED, AdaptDel, and AdaptDel+ certifies Leveshtein distance perturbations, while RanMASK only certifies Hamming distance perturbations. Thus, the actual robustness of RanMASK at the same radii is lower than that of CERT-ED, AdaptDel and AdaptDel+.

All experiments in this paper are conducted using a private cluster with Intel(R) Xeon(R) Gold 6326 CPU at 2.90GHz and NVIDIA A100 GPUs. Unless otherwise specified, we use a single GPU for all experiments.

### F.1 Standardized Cost Comparison for Certification

The certification process consists of two key stages: Monte Carlo estimation and the certification logic. In the first stage, we estimate the smoothed classifier's output probabilities for the input $x$ by making $N_{\mathrm{pred}} = 1000$ and $N_{\mathrm{cert}} = 4000$ forward calls to the base model, as shown in Lines 1–2 of Algorithm 1. This stage dominates the overall runtime. In the second stage, the certification logic (Lines 3-9) uses these estimated probabilities to find the largest certified radius. A loose bound on the asymptotic complexity of this second step is $O(r^4)$ for certified radius $r$. This follows since Algorithm 1 (Lines 3-9) performs a linear search for the radius $r$ (Line 3), contributing a factor of $O(r)$. Within this loop, it iterates through all edit combinations $\mathcal{C}(\mathsf{o}, r)$ in Line 4, of which there are $O(r^2)$. The inner bound calculations (Lines 6 and 7) are approximately linear in the number of edits, $O(r)$.

Despite this asymptotic complexity, standardized empirical tests show AdaptDel runs 20% and 372% faster than CERT-ED and RanMASK, respectively (Figure 6). Notably, we evaluate runtime efficiency using the SpamAssassin dataset—chosen specifically as the most challenging scenario due to its

| Dataset | $p_{\text{lb}}$ | $k$ | Clean Acc. | Mean CR | Wass. Dist. | Median log(CC) |
|---|---|---|---|---|---|---|
| Yelp | 50% | 65 | 67.89 | $0.16 \pm _{0.006}$ | 0.44 | 0.00 |
| | 60% | 52 | 66.84 | $0.32 \pm _{0.007}$ | 0.34 | 0.00 |
| | 70% | 39 | 65.33 | $0.41 \pm _{0.009}$ | 0.31 | 6.32 |
| | 80% | 26 | 63.13 | $0.61 \pm _{0.013}$ | 0.27 | 6.60 |
| | 90% | 13 | 56.98 | $0.99 \pm _{0.023}$ | 0.28 | 6.89 |
| | 95% | 6 | 48.40 | $1.41 \pm _{0.037}$ | 0.32 | 6.92 |
| IMDB | 50% | 115 | 94.64 | $0.34 \pm _{0.005}$ | 0.29 | 0.00 |
| | 60% | 92 | 93.15 | $0.86 \pm _{0.005}$ | 0.39 | 6.83 |
| | 70% | 69 | 91.29 | $1.01 \pm _{0.007}$ | 0.29 | 6.83 |
| | 80% | 46 | 90.59 | $1.61 \pm _{0.010}$ | 0.25 | 12.51 |
| | 90% | 23 | 87.99 | $2.86 \pm _{0.018}$ | 0.20 | 14.77 |
| | 95% | 11 | 85.25 | $4.26 \pm _{0.028}$ | 0.16 | 20.87 |
| SpamAssassin | 50% | 115 | 98.15 | $0.53 \pm _{0.013}$ | 0.45 | 0.00 |
| | 60% | 92 | 98.19 | $1.25 \pm _{0.011}$ | 0.45 | 7.02 |
| | 70% | 69 | 97.98 | $1.61 \pm _{0.019}$ | 0.34 | 7.02 |
| | 80% | 46 | 97.98 | $3.20 \pm _{0.024}$ | 0.36 | 20.22 |
| | 90% | 23 | 97.73 | $6.25 \pm _{0.053}$ | 0.30 | 38.79 |
| | 95% | 11 | 97.69 | $11.58 \pm _{0.111}$ | 0.24 | 69.84 |
| LUN | 50% | 135 | 99.32 | $0.53 \pm _{0.006}$ | 0.34 | 7.18 |
| | 60% | 108 | 99.36 | $1.25 \pm _{0.006}$ | 0.67 | 7.18 |
| | 70% | 81 | 99.10 | $1.55 \pm _{0.009}$ | 0.42 | 7.30 |
| | 80% | 54 | 98.74 | $2.96 \pm _{0.012}$ | 0.36 | 20.51 |
| | 90% | 27 | 97.88 | $5.65 \pm _{0.028}$ | 0.30 | 38.90 |
| | 95% | 13 | 96.19 | $9.64 \pm _{0.061}$ | 0.21 | 65.74 |

Table 5: Ablation study on the deletion probability lower bound $p_{\text{lb}}$ for AdaptDel, with $k = \lfloor (1 - p_{\text{lb}}) \mathbb{E}\{|\boldsymbol{x}|\} \rfloor$. All metrics are computed on the full test set. "Mean CR" refers to the mean certified radius, reported with standard error. A larger standard error indicates greater variability in certified radii across examples. We report the Wasserstein distance ("Wass. Dist.") between the standardized distributions of certified radius and input length, as a measure of how well each method adapts to variations in input length (smaller is better). The right-most column contains the median log-cardinality ($\log(\text{CC})$) of the certified region. For the Yelp dataset, the first quartile (Q1) of $\log(\text{CC})$ is reported instead, since the median certified region cardinality is zero.

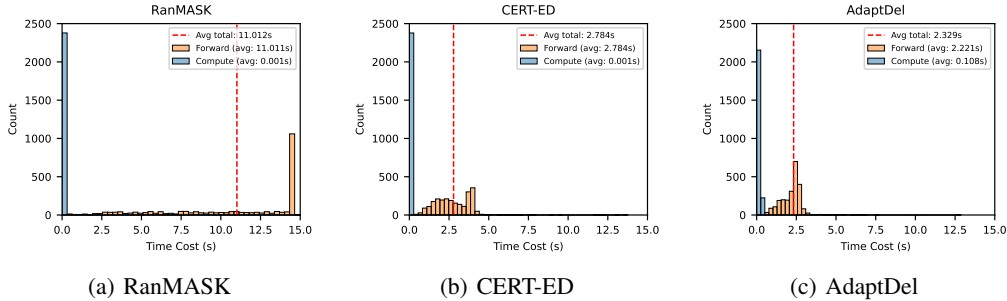

(a) RanMASK      (b) CERT-ED      (c) AdaptDel

Figure 6: From left to right are the histogram of computation cost for forward operation and computing certified radius for RanMASK, CERT-ED and AdaptDel, respectively. For this experiment, we set the batch size to 256 and the number of samples for certification to be 4096. As a result, a total of 16 batches are required to compute the certified radius. The histogram shows the distribution of the computation cost for each method. The total cost of AdaptDel is lower compared to all other methods despite the increased complexity in compute time. This is because the adaptive deletion rate provides shorter smoothed input size for longer inputs, significantly reducing the cost of the forward computation.

Table 6: Training time statistics for each dataset and model. The number of epochs varies due to early stopping.

| Model | Statistic | Dataset/Number of samples | | | |
| | | Yelp/585 000 | SpamAssassin/2 152 | IMDB/22 500 | LUN/13 416 |
| --- | --- | --- | --- | --- | --- |
| NoSmooth | epochs | 30 | 40 | 30 | 55 |
| | sec/epoch | 6 269 | 27 | 258 | 143 |
| RanMASK, 90% | epochs | 60 | 50 | 60 | 65 |
| | sec/epoch | 6 252 | 35 | 341 | 258 |
| CERT-ED, 90% | epochs | 60 | 40 | 65 | 60 |
| | sec/epoch | 2 334 | 13 | 128 | 55 |
| AdaptDel, 90% | epochs | 155 | 55 | 85 | 70 |
| | sec/epoch | 989 | 6 | 50 | 41 |
| AdaptDel+ | epochs | 90 | 65 | 140 | 75 |
| | sec/epoch | 3 146 | 14 | 73 | 83 |

| Model | Dataset/Number of samples | | | |
| | Yelp/10 000 | IMDB/25 000 | SpamAssassin/2 378 | LUN/6 454 |
| --- | --- | --- | --- | --- |
| RanMASK, 90% | 9 313 | 13 331 | 13 899 | 14 641 |
| CERT-ED, 90% | 2 633 | 3 311 | 3 319 | 4 819 |
| AdaptDel, 90% | 2 129 | 2 896 | 4 090 | 3 174 |
| AdaptDel+ | 2 469 | 1 812 | 1 857 | 2 235 |

Table 7: Certification time in milliseconds per sample on the test set for each dataset, including overheads. We use 1000 Monte Carlo samples for prediction and 4000 samples for estimating certified radii.

longer certified radii and shorter sequence lengths. Since the efficiency gains of AdaptDel are most pronounced on longer inputs where it can apply a higher deletion rate, benchmarking on a dataset with shorter sequences provides a more conservative estimate of its performance advantage. The observed efficiency improvement is primarily due to reduced forward-pass times resulting from adaptive deletion rates. In practice, computing the certified radius itself accounts for less than $5\%$ of the total runtime.

## F.2 Train

Table 6 shows the number of epochs used to train each model/dataset (with early stopping) and the training time per epoch. While AdaptDel is the fastest in training, AdaptDel+ takes similar time to train compared to CERT-ED, which is 2–3 times faster than the non-smoothed baseline, and 2–5 times faster to train than RanMASK. The total computation used across all datasets for certification is estimated to be 300 hours A100 GPU time.

In addition to the training times reported in Table 6, the AdaptDel+ method requires a one-time, offline calibration step to determine the optimal expected lengths for each bin, as detailed in Appendix C. This process is performed after the base model is trained and does not add to the online certification time. On a single NVIDIA A100 GPU, the total calibration times were: Yelp (9.5 hrs), IMDB (12 hrs), SpamAssassin (8 hrs), and LUN (12 hrs).

## F.3 Certification

Table 7 shows the average certification time per test instance, including overheads. Our approaches are upto 2 and 10 times faster than CERT-ED and RanMASK. The total computation used across all datasets for certification is estimated to be 240 hours A100 GPU time.

# G  Empirical Robustness Against Text Attacks

While the primary focus of our work is on certified robustness, we also evaluate the empirical robustness of our methods against several common text-based adversarial attacks. This analysis provides a more practical perspective on model resilience against existing attack heuristics.

**Experimental Setup**    Our experimental setup largely follows the methodology of Huang et al. [18]. We evaluate all models on a randomly selected subsample of 200 instances from each dataset's test set. For the smoothing-based defenses (AdaptDel, CERT-ED, and RanMASK), we use a Monte Carlo estimation with 100 samples to approximate the smoothed classifier's prediction. This is a smaller sample size than used for certification, a necessary compromise to make the computational cost of attacking these models feasible. To ensure all attacks terminate within a reasonable timeframe, we enforce a query budget of 3000 calls to the target model and a time budget of 2 hours per instance.

**Attack Outcomes**    Each attack on a given instance can result in one of four outcomes:

- **Success:** The attack successfully finds an adversarial example that changes the model's correct prediction to an incorrect one.
- **Failure:** The attack terminates without finding an adversarial example. This occurs if the attack exhausts its search space or reaches the predefined query limit of 3000 model evaluations.
- **Skipped:** The original input is misclassified by the model, so no attack is initiated.
- **Timeout:** The attack exceeds the time limit of 7200 seconds before terminating.

**Metrics and Baselines**    We measure performance using *robust accuracy*, defined as the fraction of non-skipped instances for which the attack outcome was either 'Failure' or 'Timeout'. This metric captures the percentage of initially correct predictions that remain correct after the attack.

We evaluate AdaptDel against several baselines: the smoothing-based methods CERT-ED [18] and RanMASK [13], and the adversarial training method FreeLB [6]. These models are tested against five representative attacks using the TextAttack framework [43], which cover a diverse range of perturbation strategies:

- **General Edit Distance Attacks:** Clare [25] and BAE-I [24] search for adversarial examples by applying a combination of word insertions, deletions, and substitutions.
- **Word Substitution Attacks:** BERT-Attack [8] and TextFooler [44] craft attacks by replacing important words with semantically similar substitutes.
- **Character-level Attack:** DeepWordBug [23] perturbs the input by applying small character-level edits (e.g., swaps, deletions) to words.

**Results and Discussion**    The robust accuracy of each model against five widely-used attacks is summarized in Table 8. We observe that AdaptDel achieves a robust accuracy that is broadly comparable to the CERT-ED baseline across most attacks, with differences often falling within the margin of statistical error for the given sample size.

This result is consistent with the primary contribution of our work. The main advantage of AdaptDel is its ability to provide stronger *certified* robustness guarantees, particularly for longer sequences, by adapting the deletion rate. This adaptation is optimized for the worst-case analysis required for certification, which does not necessarily translate to a direct advantage against the specific heuristics employed by these empirical attacks. Therefore, while AdaptDel demonstrates strong empirical resilience on par with existing methods, its key benefit remains in the domain of provable security, where it significantly expands the size of the certified region.

# H  Impact Statement

Where adversarial examples present an important threat to the integrity of learned models, randomized smoothing reflects a well-documented defense, while certifications offer a reliable measurement of robustness. Together, this research offers potential societal benefit from more resilient AI systems, and

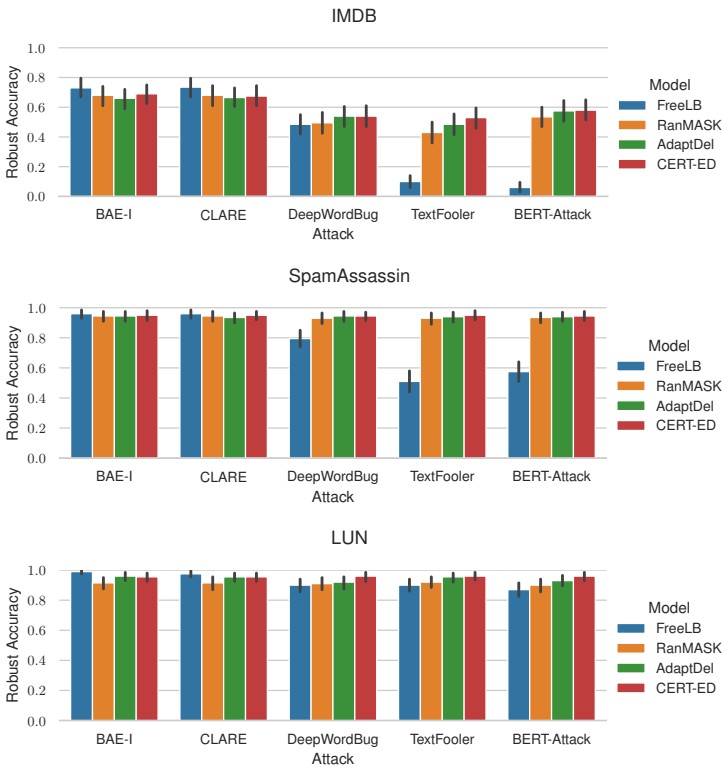

Figure 7: Robust accuracy of AdaptDel compared to baseline defenses against five common adversarial text attacks. Error bars indicate 95% bootstrap confidence intervals. Across the three datasets, AdaptDel demonstrates empirical robustness that is statistically comparable to the CERT-ED baseline. All smoothing-based methods are highly robust on the LUN and SpamAssassin datasets, while the adversarial training method FreeLB shows vulnerability to specific word-substitution attacks.

| Model | Attack | | | | |
|---|---|---|---|---|---|
| | BAE-I | BERT-Attack | CLARE | DeepWordBug | TextFooler |
| **Dataset:** IMDB | | | | | |
| AdaptDel | 0.860 / 0.660 | 0.865 / 0.575 | 0.890 / 0.665 | 0.885 / 0.540 | 0.860 / 0.485 |
| CERT-ED | 0.885 / 0.690 | 0.860 / 0.580 | 0.880 / 0.675 | 0.880 / 0.540 | 0.880 / 0.530 |
| FreeLB | 0.940 / 0.730 | 0.940 / 0.060 | 0.940 / 0.735 | 0.940 / 0.485 | 0.940 / 0.100 |
| RanMASK | 0.875 / 0.680 | 0.885 / 0.535 | 0.880 / 0.680 | 0.880 / 0.495 | 0.880 / 0.430 |
| **Dataset:** LUN | | | | | |
| AdaptDel | 0.980 / 0.960 | 0.980 / 0.930 | 0.980 / 0.955 | 0.985 / 0.920 | 0.985 / 0.955 |
| CERT-ED | 0.995 / 0.955 | 0.995 / 0.960 | 0.995 / 0.955 | 0.990 / 0.960 | 1.000 / 0.960 |
| FreeLB | 0.995 / 0.990 | 0.995 / 0.870 | 0.995 / 0.975 | 0.995 / 0.900 | 0.995 / 0.900 |
| RanMASK | 0.980 / 0.915 | 0.980 / 0.900 | 0.975 / 0.915 | 0.970 / 0.910 | 0.970 / 0.920 |
| **Dataset:** SpamAssassin | | | | | |
| AdaptDel | 0.975 / 0.945 | 0.970 / 0.940 | 0.970 / 0.935 | 0.970 / 0.945 | 0.960 / 0.940 |
| CERT-ED | 0.965 / 0.950 | 0.965 / 0.945 | 0.965 / 0.950 | 0.970 / 0.945 | 0.970 / 0.950 |
| FreeLB | 0.970 / 0.960 | 0.970 / 0.575 | 0.970 / 0.960 | 0.970 / 0.795 | 0.970 / 0.510 |
| RanMASK | 0.965 / 0.945 | 0.965 / 0.935 | 0.955 / 0.945 | 0.965 / 0.930 | 0.965 / 0.930 |

Table 8: Empirical robustness results of different models against various attacks on three datasets. Each cell shows the accuracy before and after the attack (e.g., 0.860 / 0.660 means 86.0% accuracy before the attack and 66.0% accuracy after the attack).

better transparency of limitations of robustness. Our work furthers the study of more realistic threat models than $\ell_p$-bounded perturbations found in vision, with edit distance and sequence classifiers more suitable to NLP. With longer sequences offering greater margin for smoothing by deletion without sacrificing accuracy, we are able to better balance utility and certification.

