# OpenReview forum: "AdaptDel: Adaptable Deletion Rate Randomized Smoothing for Certified Robustness"
_NeurIPS.cc/2025/Conference — NeurIPS 2025 poster_

### Official Review · Reviewer_QS5h · 2025-06-12

**Clarity:** 3
**Significance:** 3
**Originality:** 3
**Rating:** 5
**Confidence:** 2

**Summary:**

The addressed topic of this work is certified robustness for deletion-based smoothing mechanism in sequence classification against edit distance perturbations. Particularly, this work extends the fixed deletion rate in previous works to an adaptive deletion rate for discrete sequences. Theoretical guarantees are established and practical algorithms are developed.

**Questions:**

This work overall is good, and I would like to propose several issues for a discussion.

1.Existing certified defenses give rigorous theoretical guarantees but suffer from limited applicability. For example, while Gaussian noise perturbations are commonly adopted for certified robustness in image classification, such guaranteed robustness is actually insufficient against those adversarial examples. That is, the certified radius under Gaussian noises is typically much smaller than perturbation generated by basic attacks like FGSM. Regarding your deletion-based mechanism, the reviewer is curious whether your algorithm and others can be somehow effective against those more realistic discrete-input attacks (if any). Any discussions or experiments would be appreciated.

2.Missing explanations about the robustness-accuracy trade-off
As elaborated in Sec.4.1, there exists a robustness-accuracy trade-off for the proposed AdapDel and AdapDel+. In Table 4 of Appendix E.2, the clean accuracy of AdapDel and AdapDel+ is generally smaller than the compared methods. The authors are suggested to give a discussion on the reasons behind. Is it due to the intrinsic mechanism of AdapDel/AdapDel+, or the hyper-parameter settings? Could more sophisticated hyper-parameter tuning boost clean accuracy? Achieving competitive or even better clean accuracy is desirable and also further strengthens this work.

**Ethical Concerns:**

["NO or VERY MINOR ethics concerns only"]

**Final Justification:**

As I wrote in the initial review, this work is overall good and I just proposed two concerns for a discussion. In the disucssion phase, although the first concern cannot be completely addressed through experiments due to the time limit, I still acknowledge the quality of this work and increase my final rating from 4 to 5.

**Limitations:**

Yes.

**Paper Formatting Concerns:**

I have not noticed any major formatting issues in this paper.

**Quality:**

3

**Strengths And Weaknesses:**

Strengths
1.Certifications for deletion-based smoothing mechanism on discrete sequences are built.
2.There are extensive experiments, including certified robustness, hyper-parameters analysis and computational complexity.

Weakness
1.It would be appreciated that discussions or experiments on practical settings, e.g., adversarial examples on discrete inputs, are supplemented.
2.Missing explanations about the robustness-accuracy trade-off.

Refer detailed weaknesses in my questions below.

---

> ### Author Rebuttal · Authors · 2025-07-31
>
> We appreciate the reviewer's thoughtful comments and questions. We'll address each point in turn:
>
> > 1. Practical settings and adversarial examples
>
> We acknowledge that certified radii, including those in our work, may not always encompass all practical attacks. However, our approach significantly expands the certified region compared to previous methods, particularly for longer sequences. This improvement is crucial as it narrows the gap between theoretical guarantees and practical attack scenarios.
>
> It's important to note that the edit distance threat model, while not perfect, represents a significant advancement over previous work that only accommodates substitutions (Hamming distance). This allows us to certify robustness against a broader range of perturbations, including insertions and deletions, which are particularly relevant for text data [GR20, LZP+21].
>
> We also emphasize that certification isn't solely about defending against adversarial attacks. It's equally important for ensuring model stability under benign noise, which is common in real-world applications. Our method provides strong guarantees for this broader notion of robustness.
>
> > Missing explanation for robustness-accuracy trade-off
>
> We appreciate the opportunity to clarify this point. Looking at Table 4, we'd argue that the clean accuracy of AdaptDel and AdaptDel+ is generally on par with CERT-ED, with only slight drops (less than 1-2 percentage points) which may not be statistically significant.
>
> The slight variations in clean accuracy could be attributed to several factors:
>
> - Convergence of training/calibration under variable-rate deletion
>
> - The base model's ability to handle variable-rate deletion
>
> -  More aggressive deletion for medium to long sequences potentially affecting their accuracy
>
> We believe these small trade-offs in clean accuracy are well justified by the significant improvements in certified robustness, particularly for longer sequences. That said, we agree that further improving clean accuracy is desirable. This could potentially be achieved by manually adjusting the deletion rate curve for AdaptDel, or adjusting the certified accuracy threshold $\tau$ for AdaptDel+.
>
>
>
> **References**
>
> - Garg, S., and G. Ramakrishnan. “BAE: BERT-based adversarial examples for text classification.” In: EMNLP 2020.
>
> - Li, D., Y. Zhang, H. Peng, L. Chen, C. Brockett, M. Sun, and B. Dolan. “Contextualized perturbation for textual adversarial attack.” In: NACL 2021.

---

> > ### Comment · Reviewer_QS5h · 2025-08-02
> >
> > The reviewer appreciates the authors' careful rebuttal.
> >
> > Regarding my first comment concerning "*Practical settings and adversarial examples*", my core concern actually aligns directly with **Reviewer jfSu**'s question. That is, can the authors provide experiments to evaluate the robustness of the smoothed classifier against some empirical text attacks?
> >
> > I understand the authors' explanation that "*This focus on certification without empirical attack testing is the norm for work on certified robustness*", which, however, is also widely criticized for certified robustness because it gives theoretical bounds but omits empirical validation against attacks. Therefore, presenting experimental results against adversarial examples, those are generated by empirical text attacks within and beyond the certified bound, would provide a more comprehensive and in-depth evaluation on the proposed method.

---

> > > ### Author Response · Authors · 2025-08-05
> > >
> > > We appreciate the reviewer's follow up. While there is generally good support for empirical validation of CR in the community, as seen by the diversity of opinions of reviewers, we note reviewer jfSu's comment that this is not a requirement but certainly an element to be encouraged. While we can look to completing attack experiments for a camera ready, however based on our current estimate of attack times, we don’t believe these would be ready during this discussion phase. We would welcome further advice or assessment about other aspects of the paper.

---

> > > > ### Comment · Reviewer_QS5h · 2025-08-05
> > > >
> > > > Dear authors, indeed, it is an encouragement to involve experiments against practical attacks, and I do acknowledge the quality of this work. Besides, It would also be much appreciated to see such results in the camera-ready version. I will increase my rating from 4 to 5 in the final justification

---

### Official Review · Reviewer_jfSu · 2025-06-30

**Clarity:** 3
**Significance:** 3
**Originality:** 3
**Rating:** 5
**Confidence:** 4

**Summary:**

This paper focuses on a new method for certifying robustness against edit distance perturbations in sequence classification tasks using randomized smoothing. The intuition is that state-of-the-art (SOTA) solutions use a fixed-rate deletion mechanism, which has been shown to be suboptimal for real-world data where sequence lengths can vary significantly. Instead, they should use input-dependent noise when certifying robustness, as highlighted in previous work for other tasks. Thus, the paper proposes AdaptDel, a method that employs an adaptable deletion rate that dynamically adjusts based on the input length of the sequence. The paper introduces also another variant called AdaptDel+, which further optimizes this rate through input binning and empirical calibration. The experimental evaluation is comprehensive: it  includes four natural language processing datasets and shows that the proposed methods achieve up to a 30 orders of magnitude improvement in the median cardinality of the certified region and up to 50% longer certified radius. The improvements are particularly evident for longer sequences and on datasets with more uniform length variations. Finally, the paper also provides a detailed analysis of the robustness scaling with input size.

**Questions:**

How do the probabilistic guarantees of the smoothed classifier hold empirically against text attacks? (you can use the methodology used in [B]).

[B] Huang et. al., CERT-ED: Certifiably Robust Text Classification for Edit Distance, in ACL 2024.

**Ethical Concerns:**

["NO or VERY MINOR ethics concerns only"]

**Final Justification:**

I fully support the acceptance of this paper, since it is good or excellent in all its parts. The intuition of the paper is sound and novel, since using input-dependent noise to certify robustness with randomized smoothing may address better the robustness-accuracy tradeoff the smoothed classifiers. The authors have extended this intuition to sequence tasks, showing that the same intuition applies.
The proposed method to certify robustness is both interesting and useful and all the theoretical results are formally proven. Also the experimental evaluation is good and considers datasets considered in previous work detailing baselines considered in this paper. Compared to previous methods, the proposed method achieves a significantly higher certified radius and mean cardinality of the certified region. Moreover, the additional experimental results in the Appendix clearly show the greater computational efficiency of the proposed method compared to baselines.

The authors have also responded to the concern raised in my review, even though they did not provide new experimental results. However, the raised concern does not undermine the quality of the work; addressing it would have only strengthened the evidence supporting the proposed method, which is already well supported by the existing results.

**Limitations:**

yes, the authors have appropriately discussed the limitations of their work throughout the text and in an apposite section in the paper.

**Paper Formatting Concerns:**

No formatting concerns to signal.

**Quality:**

3

**Strengths And Weaknesses:**

Thank you to the authors for submitting this interesting paper. I find the research direction about improving randomized smoothing and the performance of the smoothed classifiers particularly interesting, especially when methods are based on in prior evidence. I believe the authors have done an excellent job in summarizing their findings in the main body of the paper, while the appendix contains interesting results and proofs that complete the work. Below, I discuss the strengths and weaknesses of this work, and I hope the authors can address my question.

## Strengths

The intuition of the paper is sound: as highlighted in [A], using input-dependent noise in randomized smoothing may address better the robustness-accuracy tradeoff the smoothed classifiers. The authors extend this intuition to sequence tasks, showing that the same intuition applies.
The proposed method is both interesting and useful. It is based on the formal result that allows the algorithm to compute lower and upper bounds on the smoothed class probability for a neighboring input (within the edit distance threat model). They depend only on the smoothed class probability at the original input and the size of the longest common subsequence between the two inputs. This result is then formally proven (in the Appendix) and enables computing a certificate of robustness by exploiting a reduction of the problem of computing lower and upper bounds on the probability to a knapsack problem. Subsequently, the paper introduces a length-based deletion function used in the smoothing procedure that is essential for making the work novel and relevant, since it simulates how the probability of deleting a token varies with sequence length, even though more complex deletion functions could be defined as acknowledged by the authors.

The experimental evaluation is ok and considers datasets considered in previous work detailing baselines considered in this paper. Compared to previous methods, AdaptDEL achieves a significantly higher certified radius and mean cardinality of the certified region. Moreover, the additional experimental results in the Appendix clearly show the superior performance of AdaptDEL across all datasets and its greater computational efficiency compared to baselines.
Finally, the related work section is curated and details well the differences between the proposal and close related work.

## Weaknesses

Related work describing the considered baselines, such as [B], also provides evidence of the empirical robustness of the smoothed classifier. Although this is not fundamental to show, since you have already provided results about the certified radius, it would be useful to show that the probabilistic guarantee also holds empirically by considering some text attacks, as in [B].

[A] Súkeník et. al., Intriguing properties of input-dependent randomized smoothing, in ICML 2022.

[B] Huang et. al., CERT-ED: Certifiably Robust Text Classification for Edit Distance, in ACL 2024.

---

> ### Author Rebuttal · Authors · 2025-07-31
>
> Thank you for the positive evaluation of our work. We're pleased you found our research direction interesting and our findings well-summarized. We appreciate your recognition of the sound intuition behind our approach and the comprehensive experimental evaluation.
>
> Regarding your question about empirical robustness against text attacks:
>
> We are familiar with the methodology used in [B] for testing empirical robustness. While we acknowledge the value of such empirical validation, our focus in this work is on provable robustness guarantees through certification. The certified radii we provide ensure that, with high probability, no attack within that radius can change the classifier's prediction. This means that:
>
> - Any attack operating within the certified region would be provably ineffective with high probability.
>
> - For potential attacks outside the certified region, a practical deployment of our method could employ abstention for inputs where robustness cannot be guaranteed.
>
> Given these strong theoretical guarantees, we believe our current results demonstrate the effectiveness of our approach. This focus on certification without empirical attack testing is the norm for work on certified robustness, although we appreciate the value of empirical research in adversarial ML more broadly.

---

> > ### Comment · Reviewer_jfSu · 2025-08-02
> > **Thank you to the authors for their response**
> >
> > Dear Authors,
> > thank you for your response. As I mentioned in my review, I understand that the empirical validation of your approach is not strictly necessary; it would have only added value to your work. The methodology you propose to evaluate an attack based on the certified radii provided by your method seems reasonable to me. I continue to acknowledge the quality of your work, especially given the answers you provided to the other reviewers.

---

### Official Review · Reviewer_RXHq · 2025-07-01

**Clarity:** 3
**Significance:** 3
**Originality:** 3
**Rating:** 5
**Confidence:** 4

**Summary:**

This paper extend deletion-based randomized smoothing from a fixed token deletion rate to a length adaptive rate. The authors derived a new probabilistic bound for adaptive deletion and proposed two variants AdaptDel (analytical length dependent) and AdaptDel+ (data driven bin calibration). The paper studies an interesting problem about how to model discrete certified robustness problem and make the method adaptive.

**Questions:**

See above.

**Ethical Concerns:**

["NO or VERY MINOR ethics concerns only"]

**Final Justification:**

- The authors claimed that the smoothing mechanism uses only deletion and will be more careful in the final version.
- The authors explained the key difference from lemma 4 in [1].
- The authors agreed the bound is not tight due to the constraint relaxations. However, I feel more discussion of schedule $\psi$ is lacking.
- The authors also explained the experimental results and will include details in the final version.
Overall, I feel the authors solved most of my concerns and I will recommend a score of 5.

**Limitations:**

Yes.

**Paper Formatting Concerns:**

NA.

**Quality:**

3

**Strengths And Weaknesses:**

- **Motivation**. Different from most prior work in image space, this work studies discrete text data for certified robustness. It is interesting and  well motivated by the real world needs. I feel this is meaningful to the robustness community. Can you provide some explanations with maybe a toy example of why typical continuous RS does not work?

- **Theory**. The theory part is sound with bounds for arbitrary $\psi(x)$.

(1) The operations o contains three elements del, ins, sub, but section directly jump to del only, can you briefly explain why the other two operations are dropped?

(2) For line 143, how can you make apply($\bar{x},\bar{\epsilon}$)=apply($\bar{x},\epsilon^\star$)? Is there a typo? Do you mean apply($\bar{x},\bar{\epsilon}$)=apply($x,\epsilon^\star$)?

(3) Can you explain how you define $\rho$ intuitively? Also it is not trivial to see the equivalence of lemma 3.1 from lemma4 from [1], can you elaborate a bit?

(4) If I understand correctly, the bound is derived from greedy algorithm of knapsack problem relaxation. Can you explain your bound tightness? It might be helpful to discuss it a bit for future better schedule $\psi$.

(5) Notation-wise (minor), $m$ is used to define bijection in lemma 3.1 and also define the num of trails in lemma 3.2.

- **Experiments**. The paper performed experiments on 4 diverse datasets and compared with two baselines CERT-ED and RanMASK. The computational cost is also included.

(1) In the LUN setup, AdapDel is not better than CERT-ED at small radius. And when radius is small, the improvement is also marginal in IMDB and SpamAssassin. Can you explain why?

(2) How does AdaptDel behave for extremely short inputs where deletion can harm semantics much?

[1] Huang et al. 2024, RS-Del: Edit Distance Robustness Certificates for Sequence Classifiers via Randomized Deletion

---

> ### Author Rebuttal · Authors · 2025-07-31
>
> Thank you for your thoughtful review and insightful questions. We appreciate your recognition of our work's motivation and theoretical soundness.
>
> > The operations o contains three elements del, ins, sub, but section directly jump to del only, can you briefly explain why the other two operations are dropped?
>
> Thank you for highlighting this point. We originally included a more explicit explanation of this distinction, but it was cut due to space constraints. To clarify: the edit operations in Section 2 define the edit distance we _certify_, which in our experiments includes all three operations (Levenshtein distance). Our smoothing mechanism, however, uses only deletion. This decoupling of certification and smoothing operations is a key feature of our approach, allowing us to certify robustness against a broader set of edit operations while using a simpler smoothing mechanism. We will reinstate a more explicit discussion of this in the final version to prevent any misunderstanding.
>
> > For line 143, how can you make $\mathrm{apply}(\bar{x}, \bar{\epsilon}) = \mathrm{apply} (\bar{x}, \epsilon^\star)$? Is there a typo? Do you mean $\mathrm{apply}(\bar{x}, \bar{\epsilon}) = \mathrm{apply} (x, \epsilon^\star)$?
>
> Thank you for catching this typo. The correct equation should be $\mathrm{apply}(\bar{\mathbf{x}}, \bar{\epsilon})  = \mathrm{apply} (\mathbf{x}, \epsilon)$.
>
> > Can you explain how you define $\rho$ intuitively? Also it is not trivial to see the equivalence of lemma 3.1 from lemma4 from [1], can you elaborate a bit?
>
> We define $\rho(\psi, \bar{\psi})$ purely to make the math more concise. It's a product of ratios of deletion and retention probabilities (line 145), simplifying to 1 when the deletion probability is constant ($\psi = \bar{\psi}$).
>
> Regarding Lemma 3.1, the key differences from Lemma 4 in [1] are:
>
> - We represent deletion edits as Boolean vectors aligned with $\mathbf{x}$, whereas [1] uses _sets of indices_ into $\mathbf{x}$ that are _retained_ (i.e., not deleted).
>
> - We use $\mathcal{E}(\mathbf{x}, \mathbf{\epsilon}^\star)$ to denote the edits building on $\mathbf{\epsilon}^\star$ (by flipping zeros to ones), whereas [1] uses $2^{\epsilon^\star}$ to denote the edits building on $\epsilon^\star$ (by removing indices from the set).
>
> - This representation difference explains why the direction of the inequality flips: we consider $\bar{\mathbf{\epsilon}} \sqsupseteq \bar{\mathbf{\epsilon}}^\star$, whereas [1] considers $\bar{\epsilon} \subseteq \bar{\epsilon}^\star$.
>
> - The equality relating $s(\bar{\mathbf{\epsilon}}, \bar{\mathbf{x}})$ and $s(\mathbf{\epsilon}, \mathbf{x})$ follows from (2), where an input-dependent deletion probability is used in place of the constant deletion probability from [1].
>
> > If I understand correctly, the bound is derived from greedy algorithm of knapsack problem relaxation. Can you explain your bound tightness? It might be helpful to discuss it a bit for future better schedule $\psi$.
>
> You're correct that we use a knapsack problem relaxation. The bound is sound but not tight due to constraint relaxations in the proof of Lemma 3.2 (following eq. 10).  Regarding the design of $\psi$, we believe it should be tailored based on the specific dataset and base model characteristics. For instance, some datasets may not tolerate as much noise, necessitating a more conservative $\psi$.
>
> > Notation-wise (minor), $m$ is used to define bijection in lemma 3.1 and also define the num of trails in lemma 3.2.
>
> Thanks for spotting this. We’ll use a different symbol in Lemma 3.2 to avoid confusion.
>
> > In the LUN setup, AdapDel is not better than CERT-ED at small radius. And when radius is small, the improvement is also marginal in IMDB and SpamAssassin. Can you explain why?
>
> This behavior is expected. AdaptDel is configured to match CERT-ED's deletion rate (90%) for short sequences (Fig 3a), which primarily determines the certified radius. We expect (and observe) larger certified radii for longer sequences where AdaptDel's deletion rate increases. We will incorporate this explanation in the final version of the paper.
>
>  > How does AdaptDel behave for extremely short inputs where deletion can harm semantics much?
>
> For short inputs, AdaptDel uses the same deletion rate as CERT-ED, resulting in comparable semantic impact. Our input-dependent approach is motivated by the hypothesis that longer sequences can often tolerate higher deletion rates without significant semantic loss. Our experiments on multiple datasets support this hypothesis, demonstrating the effectiveness of our approach. This empirical validation, alongside our theoretical contributions in deriving certificates for input-dependent deletion, is a core contribution of our work.
>
> Additionally, AdaptDel's increasing deletion rate with sequence length leads to more uniform post-deletion sequences. This characteristic could benefit the model across all sequence lengths by promoting more stable training and potentially easier learning of robust features, as the model encounters more consistent input lengths.

---

> > ### Comment · Reviewer_RXHq · 2025-08-05
> >
> > Thanks the authors for the detailed rebuttal. That solves my concerns. I am willing to raise my score from 4 to 5 in my final justification.

---

> > > ### Comment · Area_Chair_PePz · 2025-08-07
> > >
> > > Dear reviewer,
> > >
> > > thank you for answering to the rebuttal. Please don't forget to also send your 'mandatory acknowledgement' of the rebuttal.
> > >
> > > Best wishes
> > >
> > > AC

---

### Official Review · Reviewer_Sr1y · 2025-07-03

**Clarity:** 2
**Significance:** 3
**Originality:** 2
**Rating:** 4
**Confidence:** 3

**Summary:**

This paper proposes AdaptDel and AdaptDel+, two randomized smoothing methods for certified robustness against edit distance perturbations in sequence classification tasks. Unlike prior work that uses a fixed deletion rate (e.g., CERT-ED), AdaptDel dynamically adjusts the deletion probability based on input length, and AdaptDel+ further calibrates this adaptively via binning and empirical optimization. The authors provide a theoretical framework to support edit-distance certification with input-dependent deletion rates and demonstrate empirical improvements over existing baselines on several NLP datasets.

**Questions:**

- How computationally intensive is the AdaptDel+ calibration process, especially in scenarios with many bins or limited training data?
- Could the proposed approach be extended to certify robustness under general edit operations, including insertions and substitutions?
- Given the difficulty of interpreting the derived bounds, can the authors provide more intuition or visualization for how the adaptive deletion affects robustness?

**Ethical Concerns:**

["NO or VERY MINOR ethics concerns only"]

**Final Justification:**

- Rebuttal clarified theoretical design and empirical behavior of adaptive deletion.
- The explanation of how input-dependent smoothing improves certified robustness added important context.
- Deletion-only smoothing remains a core limitation affecting generality.
- I raise my score to 4, reflecting a more complete understanding of the method's contribution.

**Limitations:**

yes

**Quality:**

3

**Strengths And Weaknesses:**

The paper extends deletion-based randomized smoothing by introducing input-dependent deletion rates, enabling more flexible certification against edit-distance perturbations. The proposed methods, AdaptDel and AdaptDel+, are supported by new theoretical bounds and demonstrate improved certified robustness compared to prior baselines, such as CERT-ED and RanMASK, across several NLP tasks. The knapsack-based formulation and calibration strategy in AdaptDel+ are technically sound and contribute to improvements in robustness, especially for longer sequences.

However, the method is a relatively straightforward extension of CERT-ED and remains limited to perturbations that only involve deletion. The theoretical analysis, while correct, is dense and challenging to interpret, lacking an intuitive explanation. AdaptDel+ introduces additional complexity through its calibration process, yet the paper provides limited discussion on its computational cost or reproducibility. Overall, the novelty and clarity are limited, and the scope is narrow.

---

> ### Author Rebuttal · Authors · 2025-07-31
>
> We thank the reviewer for their thoughtful feedback and constructive criticism. We appreciate the recognition of our work's technical merits and the improvements in certified robustness demonstrated by our methods.
>
> > Overall, the novelty and clarity are limited, and the scope is narrow.
>
> We appreciate the reviewer's perspective. However, we respectfully disagree with this assessment. Our work introduces a novel approach to input-dependent randomized smoothing for discrete sequences, which has not been explored in previous literature. We extend the theoretical framework for randomized smoothing to variable-rate deletion, ensuring sound certification with respect to edit distance. This contribution is both novel and significant in the field of certified robustness.
>
> Regarding clarity, we would appreciate specific feedback on which aspects of the paper were unclear, so we can improve the presentation in the final version. As for the scope, our work addresses a fundamental challenge in NLP robustness, applicable to a wide range of sequence classification tasks. We believe this scope is sufficiently broad and impactful.
>
> > AdaptDel+ introduces additional complexity through its calibration process, yet the paper provides limited discussion on its computational cost or reproducibility.
>
> > How computationally intensive is the AdaptDel+ calibration process, especially in scenarios with many bins or limited training data?
>
> We thank the reviewer for highlighting this aspect.
>
> Regarding computational cost: We acknowledge that our submission did not include information about the computational cost of the AdaptDel+ calibration process. Here are the calibration times for each dataset:
>
>
> | Dataset | Calibration time (HH:MM) |
> |:---|---:|
> | Yelp | 09:34 |
> | SpamAssassin | 08:02 |
> | IMDB | 12:10 |
> | LUN | 12:06 |
>
>
>
> We note that the calibration process (Algorithm 5) has an asymptotic running time that is linear in the number of bins $n$ and sample size per bin $m$. It is done offline (following training) and does not contribute to online certification time. We will include these details in the final version of the paper.
>
> Concerning reproducibility: The process is fully detailed in pseudocode in Appendix C, including Algorithm 5. We report the parameter settings used in Appendix D.3 and provide code in the supplementary material. If there are specific aspects of reproducibility that the reviewer finds lacking, we would appreciate more detailed feedback to address these concerns.
>
> > the method is a relatively straightforward extension of CERT-ED and remains limited to perturbations that only involve deletion.
>
> > Could the proposed approach be extended to certify robustness under general edit operations, including insertions and substitutions?
>
> We appreciate the opportunity to clarify a potential misunderstanding. Our method is not limited to certifying deletion-only perturbations. Instead, it provides robustness guarantees against the generalized edit distance, where the set of edit operations $\mathsf{o}$ can be any subset of insertions, substitutions, and deletions. This is stated in Section 2, though we removed a more explicit paragraph to fit within the page limit – we will reinstate this in the final version.
>
> By contrast, our smoothing mechanism does employ deletion operations only. This decoupling of the smoothing mechanism from the certified threat model is a key feature of our approach, though it's not a new concept [LJG21, LF21]. The critical requirement is that the smoothing distribution produces statistically similar outputs for inputs that are close in edit distance, which we achieve using deletion-based smoothing.
>
> In our experiments, we focus on certification of (Levenshtein) edit distance, which is supported by our method and CERT-ED (line 243). RanMASK only supports Hamming distance certification (line 246).
>
> > Given the difficulty of interpreting the derived bounds, can the authors provide more intuition or visualization for how the adaptive deletion affects robustness?
>
> The key intuition behind our approach is that while higher deletion rates generally improve robustness, they also tend to reduce clean accuracy. However, prior work has observed that longer sequences can tolerate more aggressive deletion without as significant a drop in accuracy. Lemma 3.2 can be used to bound the stability of a smoothed classifier with input-dependent deletion rates. While the bound is complex to express mathematically, it is computationally tractable. This is not uncommon in the certification literature, where bounds often cannot be expressed analytically but must be computed numerically [LYC+19, SKG22].
>
> Our experiments demonstrate how adaptive deletion impacts robustness. In essence, we can achieve a similar robustness/accuracy trade-off for short sequences while improving performance for longer sequences. This is most clearly illustrated in Fig 2, where we plot certified accuracy grouped by quartile of input size.
>
> **References**
>
> - Liu, H., J. Jia, and N. Z. Gong. "PointGuard: Provably robust 3D point cloud classification." In: CVPR 2021.
>
> - Levine, A. and S. Feizi. "Improved, deterministic smoothing for L1 certified robustness." In: ICML 2021.
>
> - Súkenı́k, P., A. Kuvshinov, and S. Günnemann. "Intriguing Properties of Input-Dependent Randomized Smoothing." In: ICML 2022.
>
> - Lee, G. H., Y. Yuan, S. Chang, and T. Jaakkola. "Tight certificates of adversarial robustness for randomly smoothed classifiers." In: NeurIPS 2019.

---

> > ### Comment · Area_Chair_PePz · 2025-08-05
> >
> > Dear Reviewer Sr1y,
> >
> > this is a reminder to please read the author's rebuttal and respond if there are points still left to resolve. Even if not, please provide a short commend and then provide your mandatory acknowledgement.
> >
> > Thank you very much!
> >
> > Best wishes
> >
> > AC

---

> > ### Comment · Reviewer_Sr1y · 2025-08-07
> > **Response to Rebuttal**
> >
> > Thank you for the detailed response. The clarifications on the certified threat model, adaptive deletion schedule, and empirical behavior across input lengths addressed parts of my earlier concerns. The rebuttal also helped me better understand the impact of input-dependent smoothing, particularly for longer sequences.
> >
> > While the reliance on deletion-only smoothing still limits practical generality, the response has strengthened my overall assessment. I am updating my score accordingly.

---

### Comment · Area_Chair_PePz · 2025-08-02
**Reviewer - Author Discussion Period**

I wish to thank the reviewers for all their work and the authors for providing a rebuttal. We are now in the author-reviewer discussion period and I want to invite all reviewers to consider the author rebuttal and respond with points that are still open. Reviewer QS5h has kindly already kickstarted this process.

---

### Note · Authors · 2025-08-15

We sincerely thank the reviewers for their thoughtful feedback and engagement. We believe we have addressed all concerns raised during the review process, and we're encouraged that several reviewers have indicated they will increase their scores.

In response to the discussions, we commit to the following key changes in the final version:

1. Clarifying the distinction between the edits used for smoothing (deletions) and those covered by our certificate (insertions, deletions, and substitutions). This will address any potential misunderstandings about the scope of our approach.

2. Reporting the computational cost of the calibration process for AdaptDel+.

3. Including experiments examining robustness to text attacks, which are currently in progress.

We believe these changes will strengthen the paper and address the main points raised during the review process.

---

### Decision · Program_Chairs · 2025-09-17

**Decision:**

Accept (poster)

**Comment:**

I thank all authors and reviewers for their work and the helpful discussion.

**Strengths:**

- Multiple reviewers emphasized the utility of making progress on robustness against text-based attacks, moving beyond the image domain.
- In particular, the progress towards certified robustness is appreciated.
- During discussion, it could be clarified that all edits (deletions, insertions, and replacements) are protected against, even though the smoothing itself is performed via deletions only
- The edit distance is (the reviewers agreed after discussion) a relevant threat model in the text domain

**Weaknesses:**

- Multiple reviewers were concerned that empiric experiments against text attacks are missing, validating that the proposed novel smoothing method provides empiric advantages, not only advantages in terms of bounds. In other words, it may be that all bounds are just overly pessimistic and the practical robustness far exceeds the certificate. However, the reviewers agreed during reviewer-AC discussions that these kinds of experiments are not strictly required for this paper to be accepted; and the authors committed to providing the experiments for the camera-ready.
- Multiple reviewers were confused about the fact that only deletions were used and how relevant the threat model is but these concerns could be clarified.

Overall, all weaknesses of the paper could be addressed during discussion and the strengths remain substantial. As such, I wholeheartedly recommend acceptance.